# DIP1 modulates stem cell homeostasis in *Drosophila* through regulation of *sisR-1*

Jing Ting Wong[1], Farzanah Akhbar[2], Amanda Yunn Ee Ng[3], Mandy Li-Ian Tay[3], Gladys Jing En Loi[4] & Jun Wei Pek [3]

Stable intronic sequence RNAs (sisRNAs) are by-products of splicing and regulate gene expression. How sisRNAs are regulated is unclear. Here we report that a double-stranded RNA binding protein, Disco-interacting protein 1 (DIP1) regulates sisRNAs in *Drosophila*. DIP1 negatively regulates the abundance of *sisR-1* and INE-1 sisRNAs. Fine-tuning of *sisR-1* by DIP1 is important to maintain female germline stem cell homeostasis by modulating germline stem cell differentiation and niche adhesion. *Drosophila* DIP1 localizes to a nuclear body (satellite body) and associates with the fourth chromosome, which contains a very high density of INE-1 transposable element sequences that are processed into sisRNAs. DIP1 presumably acts outside the satellite bodies to regulate *sisR-1*, which is not on the fourth chromosome. Thus, our study identifies DIP1 as a sisRNA regulatory protein that controls germline stem cell self-renewal in *Drosophila*.

[1] Ngee Ann Polytechnic, 535 Clementi Road, Singapore 599489, Singapore. [2] Temasek Polytechnic, 21 Tampines Avenue 1, Singapore 529757, Singapore. [3] Temasek Life Sciences Laboratory, 1 Research Link National University of Singapore, Singapore 117604, Singapore. [4] National University of Singapore, 21 Lower Kent Ridge Road, Singapore 119077, Singapore. Jing Ting Wong and Farzanah Akhbar contributed equally to this work. Correspondence and requests for materials should be addressed to J.W.P. (email: junwei@tll.org.sg)

Recent studies have uncovered a class of stable intronic sequence RNAs (sisRNAs) that are derived from the introns post splicing[1]. sisRNAs are present in various organisms such as viruses, yeast, *Drosophila*, *Xenopus*, and mammals[1–11]. Studies in *Drosophila* and mammalian cells suggest that sisRNAs function in regulating the expression of their parental genes (host genes where they are derived from) via positive or negative feedback loops[5, 10, 11]. In yeast, sisRNAs are involved in promoting robustness in response to stress[7], while in *Drosophila*, sisRNAs have been shown to be important for embryonic development[12]. However, very little is understood about the biological functions of sisRNAs in terms of regulating cellular processes such as differentiation, proliferation, and cell death.

The *Drosophila* genome encodes for several double-stranded RNA (dsRNA) binding proteins that localize to the nucleus[13, 14]. Most of them have been found to regulate specific RNA-mediated processes such as RNA editing, X chromosome activation, and miRNA biogenesis[15–20]. The Disco-interacting protein 1 (DIP1) is a relatively less characterized dsRNA binding protein that has been implicated in anti-viral defense and localizes to the nucleus as speckles. Otherwise, not much is known about the biological processes regulated by DIP1 [21–24].

In this paper, we show that the regulation of a *Drosophila* sisRNA sisR-1 by DIP1 is important for keeping female germline stem cell homeostasis in place. We also show that DIP1 regulates INE-1 sisRNAs and localizes to a previously undescribed nuclear body around the fourth chromosomes, called the satellite body. The regulation of *sisR-1*, which is not on the fourth chromosome, by DIP1 presumably does not occur in the satellite bodies.

## Results

**DIP1 regulates *sisR-1*.** We previously identified a sisRNA sisR-1 in *Drosophila*[5]. To identify proteins that regulate sisR-1

abundance, we employed a candidate approach of dsRNA binding proteins that localize to the nucleus[13]. We began by searching for candidate genes that are specific to the sisRNA pathway. In *Drosophila*, there are seven dsRNA binding proteins that localize to the nucleus[13]. Over the years, many of the original dsRNA binding proteins have been assigned functions to the miRNA and RNAi pathways. For example, CG188 (now *pasha*) and *drosha*, and CG12493 (now *blanks*) have been assigned to the miRNA and siRNA pathways respectively[25, 26]. CG12598 (now *adar*) and *mle* have been shown to be required for RNA editing and X-activation respectively[20, 27]. Among the seven, DIP1 and CG8273 are the least characterized and therefore most promising. The role of CG8273 in *sisR-1* regulation will be reported in a separate study. We found that DIP1 functions to reduce the levels of *sisR-1*. DIP1 is a dsRNA binding protein that localizes to nuclear speckles and regulates cell fate decisions during development[23]. In *DIP1* mutant ovaries, *sisR-1* level was upregulated (Fig. 1a and Supplementary Fig. 1a), suggesting that *DIP1* negatively regulates *sisR-1*. Since the *rga* mRNA was downregulated in *DIP1* mutant ovaries (Supplementary Fig. 1b), the upregulation of *sisR-1* was not due to increase in *rga* transcription. Conversely, germline overexpression of *DIP1* using a transgenic fly containing an EP element upstream of the 5′UTR of the *DIP1* locus resulted in a decrease in the abundance of *sisR-1* (Fig. 1a and Supplementary Fig. 1a). These results indicated that *DIP1* represses *sisR-1* abundance. However, we were not able to perform in situ hybridization to detect *sisR-1* in the ovaries due to its low abundance and inability to distinguish *sisR-1* from *rga* pre-mRNA.

We next examined whether *DIP1* affects the expression of *sisR-1* via a transcriptional or post-transcriptional manner. We treated wild-type and *DIP1* overexpression ovaries with actinomycin D to inhibit transcription and observed any changes in *sisR-1* abundance. Previous analysis showed that a 30 min treatment was sufficient to inhibit transcription in the ovaries[5].

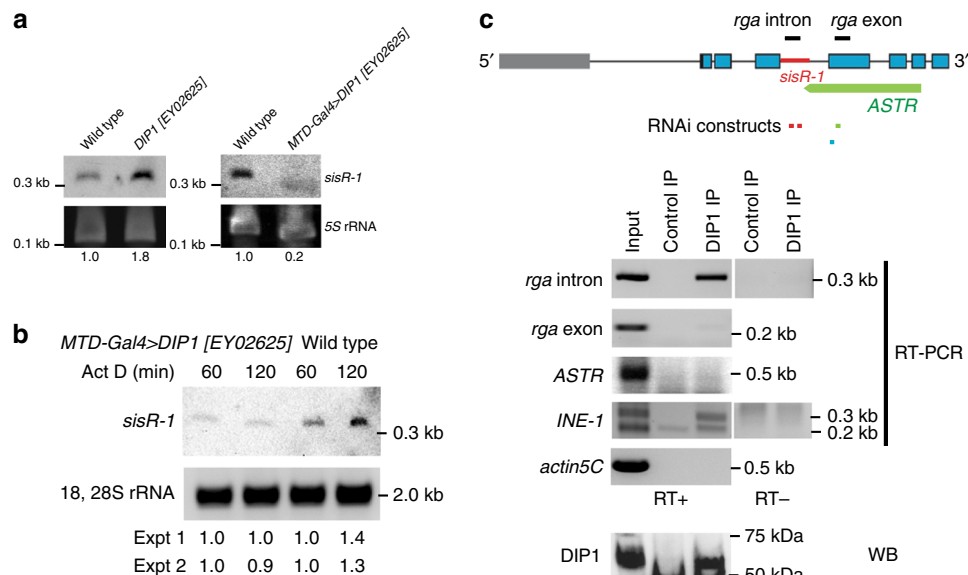

**Fig. 1** DIP1 regulates *sisR-1*. **a** Representative northern blots (out of at least three biological replicates) showing the expression of *sisR-1* in the ovaries of the indicated genotypes. 5S rRNA was used as a loading control. Numbers below indicate relative band intensities of *sisR-1* normalized to rRNA quantified using ImageJ software. **b** Northern blot showing the levels of *sisR-1* in the ovaries of the indicated genotypes after 60 and 120 min of actinomycin D treatment. 18, 28S rRNA was used as a loading control. Numbers below indicate relative band intensities (120 min compared to 60 min) of *sisR-1* normalized to rRNA quantified using ImageJ software from two independent experiments. **c** RT-PCR and western blot showing enrichment of *sisR-1* and INE-1 but not *rga* exonic sequences and *ASTR* in DIP1 immunoprecipitated samples. *Actin5C* was used as a negative control for unspecific pull down. The regions where the intronic and exonic sequences were amplified by PCR were shown on the *rga* gene model. The *red line* in the intron is where *sisR-1* is encoded. Locations of RNAi constructs used to target *sisR-1* (*red*), *ASTR* (*green*), and *rga* (*blue*) were indicated. WB, western blot

In control ovaries, we observed a strong accumulation of *sisR-1* over time from 60 to 120 min post treatment (Fig. 1b), indicating that mature and stable *sisR-1* was being produced from the precursors during this period of time. However, in ovaries overexpressing *DIP1*, the accumulation of *sisR-1* was perturbed (Fig. 1b). No observable change was seen for 18, 28S rRNA (Fig. 1b). This observation suggests that *DIP1* may regulate the stability and/or processing of *sisR-1* in a post-transcriptional manner. It is interesting to note that a previous study had shown that actinomycin D can inhibit the turnover of a nuclear form of *hsr-omega* ncRNA in *Drosophila*[28]. Together with our observation, it implies that actinomycin D may regulate the decay of specific ncRNAs in the nucleus by a yet unknown mechanism.

To ask whether the effect of DIP1 on *sisR-1* is direct, we checked whether DIP1 interacts with *sisR-1* in a complex by performing in vivo co-immunoprecipitation. We generated an antibody against DIP1 (Supplementary Fig. 1c) and immunoprecipitated endogenous DIP1 in S2 cells (Fig. 1c). RT-PCR revealed that DIP1 interacted with *sisR-1*, but not with the exonic sequences of *rga* and *actin5C* as negative controls (Fig. 1c), suggesting that DIP1 directly regulates the stability of *sisR-1*. Moreover, DIP1 did not interact with *ASTR*, a target of *sisR-1*, suggesting that DIP1 is not part of the *sisR-1* silencing complex (Fig. 1c). Thus, DIP1 binds to *sisR-1* and regulates steady-state *sisR-1* levels in *Drosophila*.

**sisR-1 promotes GSC–niche occupancy**. In *Drosophila*, *sisR-1* had been shown to regulate the expression of its host gene *regena* (*rga*)[5]. The *rga* gene had been identified as a candidate gene that regulates germline stem cell (GSC) differentiation in the ovaries[29]. To assess the biological significance of DIP1-mediated regulation of *sisR-1*, we therefore examined if *sisR-1* regulates GSC self-renewal or differentiation. In the *Drosophila* germarium, GSCs can be identified by their location in the niche (anterior tip) and the stereotypic location of spherical spectromes at the GSC–niche interface (hereafter referred as GSC–niche occupancy, which means GSCs that are in physical contact with the niche, Fig. 2a, b). We found that manipulating the expression of *sisR-1* using previously reported RNAi lines, which specifically knocked down *sisR-1* but not its parental *rga* gene, changed the GSC–niche occupancy[5] (see "Methods"). Immunostaining showed a decrease in the number of GSCs in two independent *sisR-1* RNAi flies compared to controls at day 7 after eclosure (appearance of germaria with 0 GSCs not found in controls, Fig. 2b–e), suggesting that *sisR-1* is required for the maintenance of GSCs in the niche.

Previously, *sisR-1* was shown to repress the expression of *ASTR*, possibly via its 3′ tail (Supplementary Fig. 2a)[5]. To test if the 3′ end of *sisR-1* is required for silencing of *ASTR* in vivo, we overexpressed wild-type and mutant forms of *sisR-1*, and assayed for their ability to repress *ASTR*. Both m1 and m2 versions of *sisR-1* were mutations introduced at different regions of the 3′ tail to disrupt complementary base pairing to *ASTR* (Supplementary Fig. 2b). Transgene expression was verified by measuring *dsRed* expression (Supplementary Fig. 2c). We observed that *MTD-Gal4*-induced overexpression of wild-type *sisR-1* was indeed more effective than m1 and m2 in repressing *ASTR* in the ovaries (Supplementary Fig. 2c), suggesting that the 3′ end of *sisR-1* is required for target suppression. Furthermore, overexpression of *sisR-1* caused an increase in the number of GSCs (appearance of germaria with four GSCs, Fig. 2f–h), confirming that *sisR-1* is sufficient to promote GSC–niche occupancy cell autonomously. However, no significant increase in GSCs was observed when we overexpressed *sisR-1* containing mutations at the 3′ ends (Fig. 2h and Supplementary Fig. 2d, e), indicating that the 3′ end is required for its function.

We next examined the cellular mechanisms by which *sisR-1* promotes GSC maintenance. Firstly, we did not observe any increase in cell death in *sisR-1* RNAi GSCs (Supplementary Fig. 2f–h), indicating that GSC loss was not due to apoptosis. GSCs are maintained by adhesion to the cap cells via E-Cadherin, and adhesion molecules are essential for niche occupancy during stem cell competition[30–34]. We observed that the expression of E-Cadherin between GSCs and cap cells was frequently low in *sisR-1* RNAi ovaries (Fig. 2i–l, *arrowheads*), suggesting that GSCs may be lost due to poor adhesion to the niche. Indeed, overexpression of E-Cadherin suppressed the GSC loss phenotype. (Fig. 2q and Supplementary Fig. 2k–n). Egg chambers with overexpression of *sisR-1* had higher levels of E-Cadherin compared to controls (Supplementary Fig. 2i, j), indicating that *sisR-1* regulates germline-soma cell adhesion. However, a previous report showed that overexpression of E-Cadherin alone is not sufficient to increase GSC number[35], suggesting that *sisR-1* regulates GSCs via cell adhesion and additional mechanisms.

Overexpression of a Trim-NHL protein Mei-P26 in GSCs can lead to precocious differentiation[36, 37]. Mei-P26 expression is low in GSCs and increases upon differentiation (Fig. 2m) and therefore needs to be carefully regulated[37, 38]. In *sisR-1* RNAi ovaries, GSCs were frequently seen to upregulated Mei-P26, which was never seen in controls (Fig. 2m–p, *arrowheads*). To investigate if GSC differentiation in *sisR-1* RNAi ovaries was due to upregulation of Mei-P26, we remove a copy of *mei-P26*. This experiment resulted in a suppression of the *sisR-1* RNAi phenotype (Fig. 2q and Supplementary Fig. 2o, p). In addition, we did not observe any defects on the *decapentaplegic* pathway and Bag-of-marbles expression (Supplementary Fig. 2q–ab). Together, our data reveal that *sisR-1* promotes GSC–niche occupancy via at least two genetic pathways: (1) promoting GSC–niche association via E-Cadherin and (2) repression of a differentiation factor Mei-P26.

**sisR-1 regulates GSC–niche occupancy via ASTR and rga**. In *Drosophila*, *sisR-1* regulates *rga* via a negative feedback loop by repressing a cis-natural antisense transcript called *ASTR* (Fig. 3a)[5]. It was shown previously that *sisR-1* represses the abundance of *ASTR*, while *ASTR* functions as a positive regulator of *rga* expression. Therefore, *ASTR* functions as an intermediate between *sisR-1* and *rga* in a negative feedback mechanism. Expression of *rga* and *ASTR* was detected during the early stages of oogenesis suggesting an active *sisR-1* axis (Supplementary Fig. 3a–c). Since *sisR-1* represses *ASTR*, we expected that *ASTR* RNAi flies to mimic the phenotype of *sisR-1* overexpression flies. As expected, knockdown of *ASTR* resulted in an increase in GSC–niche occupancy (Fig. 3b, j). Overexpression of *ASTR* led to a significant change in distribution of germaria to having less GSCs (Supplementary Fig. 3d, e), confirming that *ASTR* limits GSC–niche occupancy. Importantly, the GSC loss phenotype in *sisR-1* RNAi was rescued by *ASTR* knockdown (Fig. 3c, d, j), in agreement with the model that *ASTR* is downstream of *sisR-1* in the regulation of GSC–niche occupancy. Since the *ASTR* RNAi phenotype was significantly altered by *sisR-1* RNAi, it also suggests that *sisR-1* may have additional targets or the *ASTR* RNAi was partial.

Next, we observed an increase in GSC–niche occupancy in *rga* RNAi flies (Fig. 3e, j and Supplementary Fig. 3f). Reciprocally, overexpression of Rga resulted in a decrease in GSCs (Fig. 3f, j and Supplementary Fig. 3g). Knockdown of *rga* suppressed the GSC loss phenotype in *sisR-1* RNAi flies (Fig. 3g, h, j). This result is consistent with the pathway that *sisR-1* represses *rga*. Finally, overexpression of *rga* suppressed the *ASTR* RNAi phenotype (Fig. 3i, j), placing *rga* downstream of *ASTR*. Taken together, our

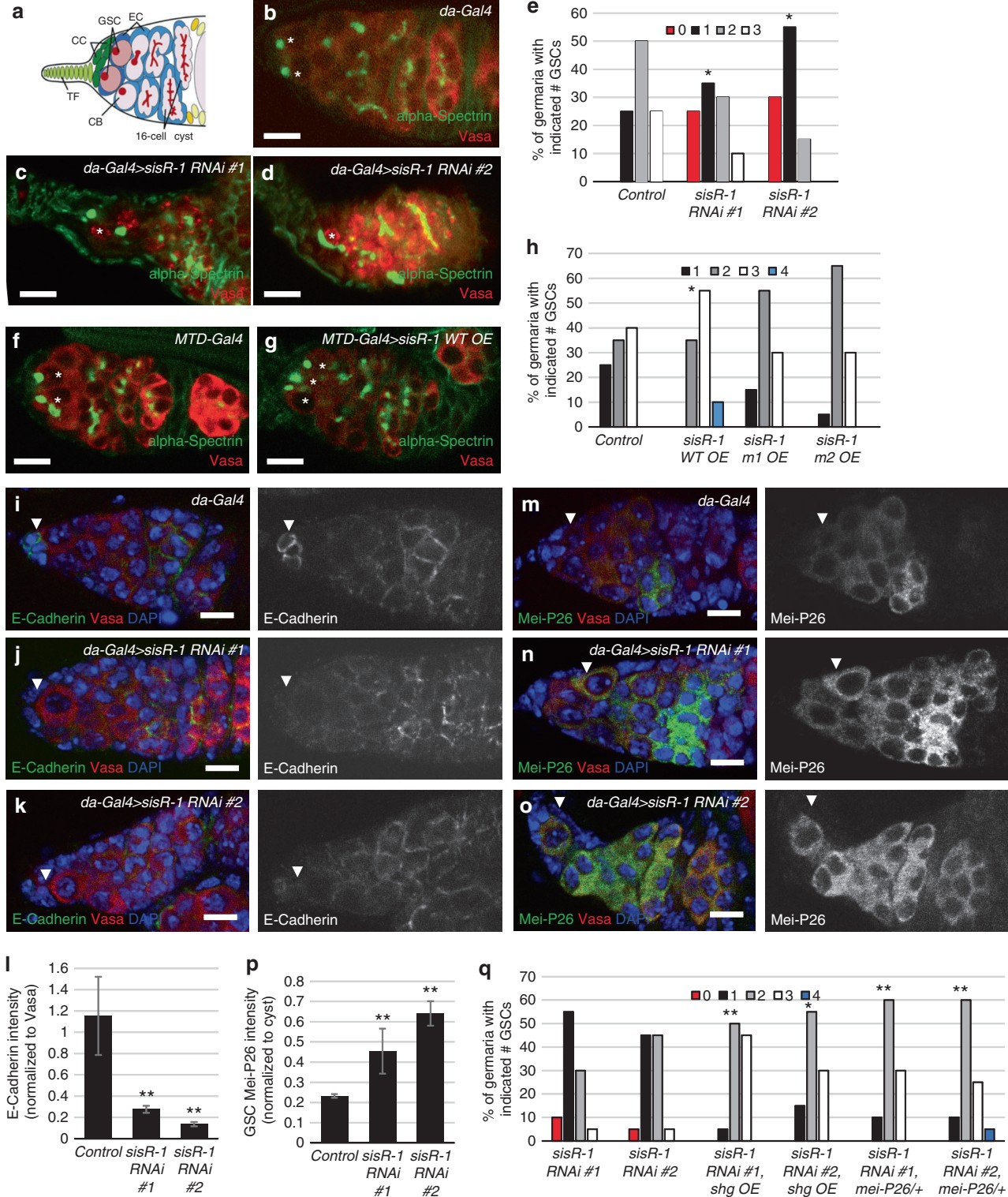

**Fig. 2** s*isR-1* regulates GSC–niche occupancy by maintaining GSC–niche adhesion and suppressing GSC differentiation. **a** A drawing of a germarium. GSC, germline stem cell, CC, cap cell, TF, terminal filament, EC, escort cell, CB, cystoblast. **b–d**, **f–g** Confocal images of germaria of the indicated genotypes stained for alpha-Spectrin (*green*) and Vasa (*red*). *Scale bar*: 10 μM. *Asterisks*(*) mark the GSCs. **e**, **h**, **q** Charts showing the percentages of germaria with the indicated number of GSCs of the indicated genotypes. *P < 0.05, **P < 0.01. Fisher's exact test. N = 20. **i–k** Confocal images of germaria of the indicated genotypes stained for E-Cadherin (*green*), Vasa (*red*), and DAPI (*blue*). *Arrowheads* point to GSC–niche interfaces. **m–o** Confocal images of germaria of the indicated genotypes stained for Mei-P26 (*green*), Vasa (*red*), and DAPI (*blue*). *Scale bar*, 10 μM. *Arrowheads* point to GSCs. **l**, **p** Charts showing the quantification of E-Cadherin and Mei-P26 intensities. **P < 0.01. N = 3. *Error bars* represent s.d.

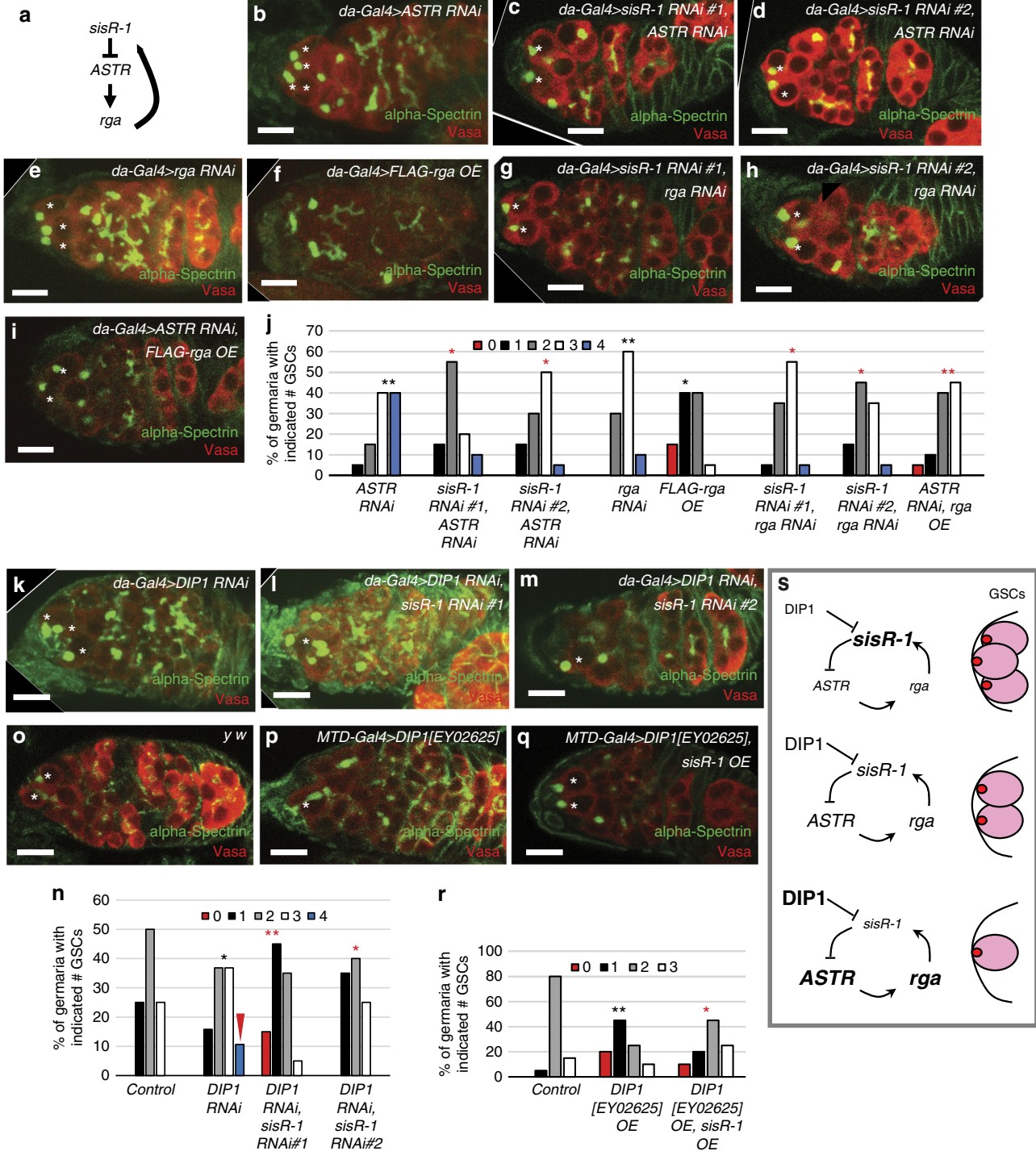

**Fig. 3** *DIP1* regulates GSC–niche occupancy by repressing *sisR-1*. **a** The *sisR-1/ASTR/rga* genetic pathway. **b–i**, **k–m**, **o–q** Confocal images of germaria of the indicated genotypes stained for alpha-Spectrin (*green*) and Vasa (*red*). *Scale bar*: 10 μM. *Asterisks* (*) mark the GSCs. **j**, **n**, **r** Charts showing the percentages of germaria with the indicated number of GSCs of the indicated genotypes. *P < 0.05, **P < 0.01. Fisher's exact test (**j**, **r**). Wilcoxon test (**n**). N = 20. *Black asterisks* are used when compared to control, *red asterisks* are used when compared to the respective RNAi or overexpression lines (*sisR-1 RNAi, ASTR* RNAi. *DIP1* RNAi and *DIP1* OE). *Red arrowhead* in **n** indicates the appearance of germaria with four GSCs that are very rarely seen in controls. **s** A model for the regulation of GSC–niche occupancy by *sisR-1*

genetic analyses established a role for the *sisR-1* axis in regulating GSC–niche occupancy in the *Drosophila* ovaries.

**DIP1 regulates GSC number by repressing *sisR-1*.** Having found that *sisR-1* regulates GSC self-renewal via *ASTR* and *rga*, we further investigated if DIP1 functions to regulate GSC–niche

occupancy by modulating the levels of *sisR-1*. Knockdown of *DIP1* led to an increase in the percentage of germaria having > 2 GSCs (Fig. 3k, n). This result was confirmed in the *DIP1* mutant, which also showed an increase in germaria having > 2 GSCs (Supplementary Fig. 1d, *arrowhead*). Furthermore, over-expression of *DIP1* using *MTD-Gal4* resulted in a significant

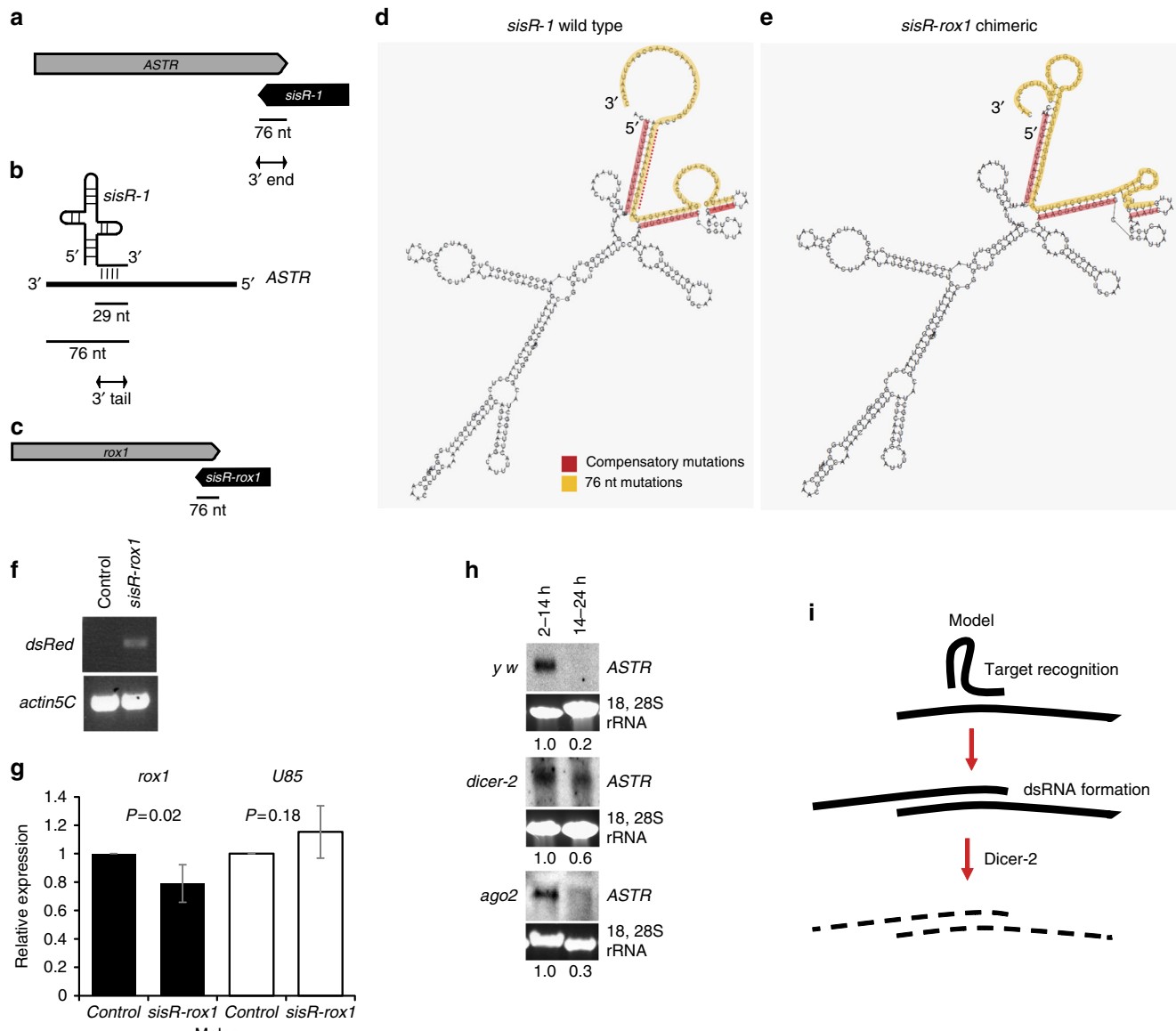

**Fig. 4** A model for *sisR-1*-mediated silencing of *ASTR*. **a** Drawing showing that *sisR-1* and *ASTR* can form a 76 nt dsRNA at the 3′ ends. Not drawn to scale. **b** Drawing showing the targeting of *ASTR* by the 3′ tail (29 nt) in *sisR-1*. **c** Drawing showing that *sisR-rox1* and *rox1* can form a 76 nt dsRNA at the 3′ ends. Not drawn to scale. **d**, **e** Predicted secondary structures of **d** wild-type *sisR-1* and **e** chimeric *sisR-rox1*. *Orange*: region of 76 nt that base pairs with *rox1*. *Red*: compensatory mutations introduced to maintain the secondary structure. **f** RT-PCR showing the expression of *sisR-rox1* transgene in male flies. **g** RT-qPCR showing the downregulation of *rox1* (*black bars*) but not *U85* (*white bars*) in control versus *sisR-rox1* expressing male flies. N = 3–4 biological replicates. Student's *t*-test was performed. *Error bars* represent s.d. **h** Northern blots (from at least two biological replicates) showing the relative abundance of *ASTR* between 2–14 h and 14–24 h embryos in *y w* control, *dicer-2*[L811fsX] and *ago*[414] mutants. 18, 28S rRNA was used as loading controls. Numbers below indicate relative band intensities of *ASTR* normalized to rRNA quantified using ImageJ software. **i** A possible model of how *sisR-1* may mediate decay of *ASTR*

decrease in GSCs (Fig. 3o, p, r). To determine if *DIP1* regulates GSCs through *sisR-1*, we knocked down *sisR-1* expression in *DIP1* RNAi flies. In this situation, the GSC increase phenotype seen in *DIP1* RNAi flies was indeed suppressed (Fig. 3l–n). We further asked if overexpression of *sisR-1* could suppress the *DIP1* over-expression phenotype. In flies that overexpressed both *sisR-1* and *DIP1* under the control of *MTD-Gal4*, the number of GSCs was reverted to wild-type level (Fig. 3q, r). Together, our results put forward a model that *DIP1* limits the steady-state levels of *sisR-1* to maintain GSC homeostasis in the niche (Fig. 3s).

**A model for *sisR-1*-mediated silencing.** Next, we attempted to gain some insights to the mechanism of *sisR-1*-mediated *ASTR*

silencing. We refer "3′ end" as the region of *sisR-1* that overlaps with the 3′ end of *ASTR* by 76 nucleotides (nt) (Fig. 4a), while "3′ tail" refers to the 29 nt region that binds to *ASTR* based on its predicted secondary structure (Fig. 4b)[5]. We have shown that the 3′ tail of *sisR-1* is necessary for silencing of *ASTR* (Supplementary Fig. 2a–c). To test if the 3′ tail is sufficient for target degradation, we replaced the 3′ tail with an antisense element that base pairs with a nuclear long ncRNA *rox2* (Supplementary Fig. 4a, b). Ectopic expression of *sisR-rox2* in S2 cells and in vivo did not lead to a significant downregulation of endogenous *rox2* (Supplementary Fig. 4c, d), indicating that the 3′ tail is not sufficient for target degradation. Taken together, our data suggest that the 3′ tail of *sisR-1* is necessary but not sufficient for target repression.

Next, we tested whether complementary base pairing between the 76 nt 3′ end of *sisR-1* with the target is sufficient for target degradation. We replaced the 3′ end of *sisR-1* with a 76 nt antisense element against an endogenous long ncRNA *rox1* to design a chimeric *sisR-1* (named *sisR-rox1*) (Fig. 4c–e). Compensatory mutations were also introduced to the regions that base pair with the new 76 nt end sequences to preserve the secondary structure as much as possible (Fig. 4d, e, *red*: compensatory mutations, *orange*: 76 nt mutations). When ectopically expressed in the adult males (Fig. 4f), *sisR-rox1* could downregulate the expression of *rox1* (Fig. 4g). Although the fold-change in *rox1* level was modest, it was statistically significant ($P$ value = 0.02). As endogenous expression of *rox1* was high and comparable to housekeeping gene *actin5C*, a ~20% change was considerable in terms of copy number. Furthermore, the effect was specific to *rox1* as another nuclear long ncRNA *U85* remained unchanged in *sisR-rox1* expressing flies (Fig. 4g).

Dicer-2 (Dcr-2) has been shown to localize to and function in the nucleus in *Drosophila* and human cells[39–42]. In vitro studies have shown that a ~40 nt duplex is sufficient for Dcr-2-mediated cleavage[43]. In wild types, endogenous *ASTR* is highly expressed in the early embryos (2–14 h) and dramatically downregulated in the late embryos (14–24 h) (Fig. 4h)[5]. In *dcr-2* mutants, the downregulation of *ASTR* is severely perturbed in late embryos (Fig. 4h), consistent with a requirement for Dcr-2 (Fig. 4h). Since Dcr-2 is a component of the endogenous siRNA pathway, we asked if the downstream effector Argonaute2 (Ago2) is also required for *ASTR* silencing. In *ago2* mutants, we did not observe any defects in the downregulation of *ASTR* in the late embryos (Fig. 4h), further suggesting that siRNAs are not involved. This observation is very similar to what was reported in human and mice where Dicer1 regulates Alu RNA independent of its RNA interference function[40]. Our results are consistent with a model in which a naturally occurring sisRNA *sisR-1* represses its target via two sequential steps: (1) target recognition via complementary base pairing of 3′ tail to its target, and (2) more extensive dsRNA formation between *sisR-1* and its target, which is cleaved by endonuclease Dcr-2 (Fig. 4i).

**DIP1 localizes as foci in the nuclei of the germline cells.** Many proteins that are involved in RNA metabolism localize to specific nuclear or cytoplasmic bodies such as Cajal bodies, Histone Locus Bodies (HLBs), Processing bodies, nuage, and Yb bodies[44–49]. To investigate the cellular localization of DIP1, we performed immunostaining using our DIP1 antibody. DIP1 localizes as nuclear foci in germline cells in the germaria (Supplementary Fig. 1e), similar to what was previously reported[23]. The staining was dramatically reduced in DIP1 mutant ovaries (Supplementary Fig. 1f). In the transcriptionally active nurse cells in the stage 5 egg chamber, DIP1 also localizes as discrete nuclear foci (Supplementary Fig. 1g, h). However, DIP1 does not form nuclear foci in the transcriptionally quiescent oocyte nucleus and cyst cells during mitosis (Supplementary Fig. 1i, j), suggesting that formation or maintenance of DIP1 nuclear foci may be associated with active transcription. In the testes, DIP1 also forms nuclear foci in the primary spermatocytes. Discrete nuclear foci could be seen during the transition from spermatogonia to primary spermatocytes (Supplementary Fig. 1k, l). When the primary spermatocytes grow larger with massive transcription of fertility genes, DIP1 staining filled the entire nuclei (Supplementary Fig. 1m). Furthermore, DIP1 staining was not detected in the sperm bundles where no transcription occurs (Supplementary Fig. 1n). Taken together, our data indicated that DIP1 localizes as nuclear foci in transcriptionally active germline cells.

We further generated HA-tagged DIP1 transgenic flies (Supplementary Fig. 1o, p). When expressed in germline cells,

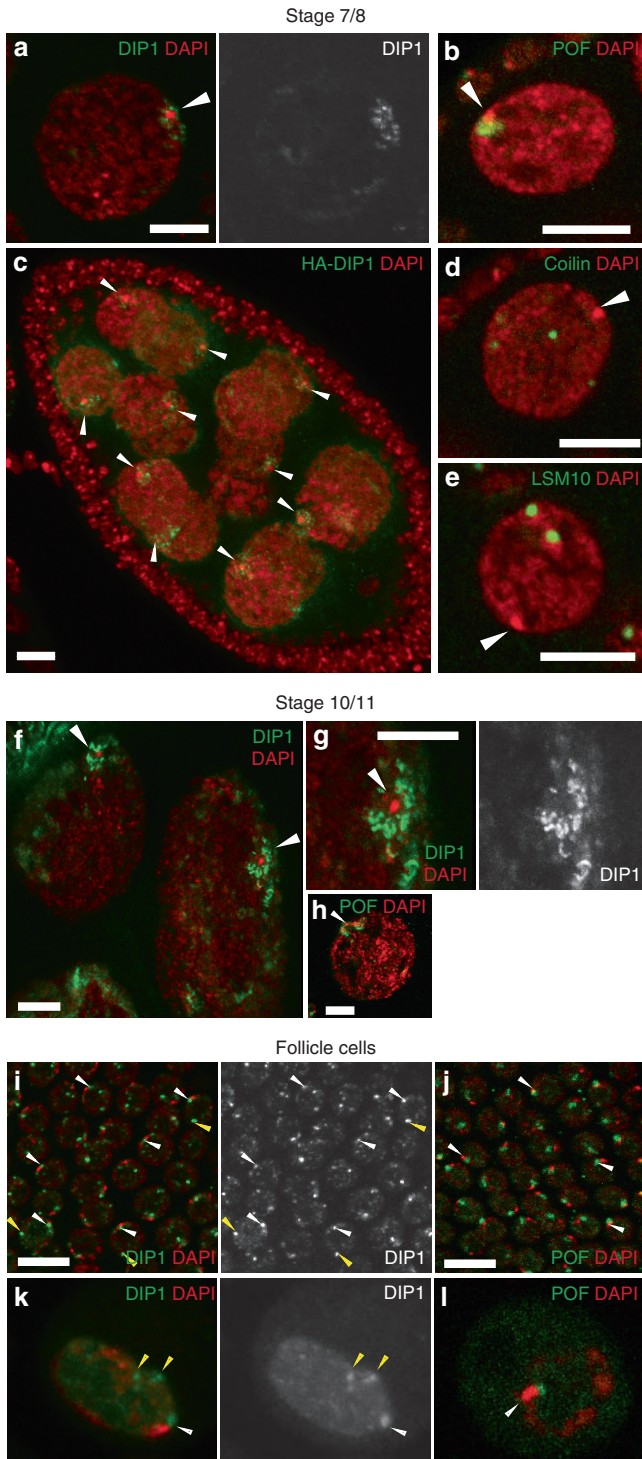

**Fig. 5** Satellite bodies are DIP1-positive nuclear bodies that decorate the fourth chromosomes. **a** A stage 7/8 nurse cell stained for DIP1 and DAPI. **b**, **d**, **e** A stage 7/8 nurse cell stained for **b** POF, **d** Coilin, and **e** LSM10. **c** A stage 7/8 egg chamber expressing HA-DIP1 protein under *MTD-Gal4*, stained with anti-HA. **f** Two stage 10/11 nurse cells stained for DIP1 and DAPI. **g** A magnified view of satellite bodies in **f**. **h** A stage 10/11 nurse cell stained for POF and DAPI. *Arrowheads* point to DAPI-dense regions of fourth chromosomes. **i**, **j** Follicle cells stained for **i** DIP1 and **j** POF. **k**, **l** S2 cells stained for **k** DIP1 and **l** POF. *White arrowheads* point to chromosome four associated bodies, while *yellow arrowheads* point to bodies that do not associate with chromosome four. *Scale bar*: 10 μm

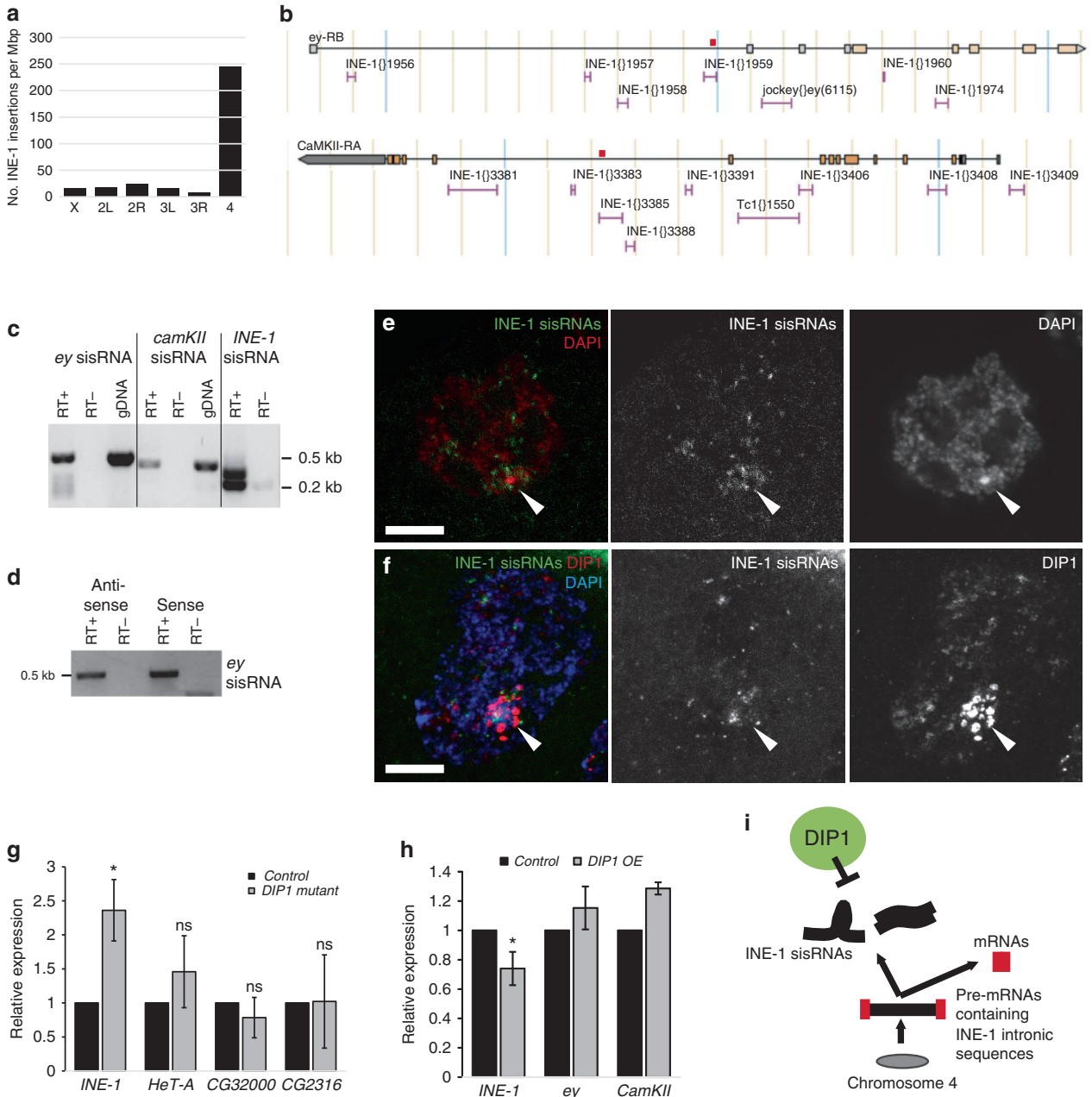

**Fig. 6** DIP1 suppresses the level of INE-1 sisRNAs. **a** Chart showing the number of INE-1 insertions per Mbp of DNA on the different chromosomes in *Drosophila melanogaster*. **b** Genetic loci for *eyeless* (*ey*) and *CamKII* showing the location of INE-1 insertions in the introns. *Red bars* indicate the gene-specific INE-1 sisRNAs amplified in **c**. **c** RT-PCR showing the presence of *ey*, *CamKII* and INE-1 sisRNAs in unfertilized eggs. gDNA, genomic DNA. **d** Strand-specific RT-PCR showing the presence of both anti-sense and sense *ey* sisRNAs in unfertilized eggs. **e** Confocal images showing the localization of INE-1 sisRNAs (*Green*) and DAPI (*Red*) in a nurse cell nucleus. *Arrowheads* point to DAPI-dense regions of fourth chromosomes. **f** Confocal images showing the localization of INE-1 sisRNAs (*Green*), DIP1 (*Red*), and DAPI (*Blue*) in a nurse cell nucleus. *Arrowheads* point to DAPI-dense regions of fourth chromosomes. **g** RT-qPCR showing the relative levels of INE-1, *HeT-A*, *CG32000*, and *CG2316* in control and *DIP1* mutant ovaries. **h** RT-qPCR showing the relative levels of INE-1, *ey* and *CamKII* in *MTD-Gal4* control and *MTD-Gal4 > HA-DIP1* ovaries. *$P < 0.01$. Two-tailed *t*-test. $N = 3$ independent biological replicates. *Error bars* represent s.d. **i** Working model on the role of DIP1-positive satellite bodies in regulating the abundance of INE-1 sisRNAs in *Drosophila*

HA-DIP1 exhibited the same nuclear foci that were detected by DIP1 antibody (Supplementary Fig. 1q, r). To test if RNA is required for the maintenance of DIP1 nuclear foci, we treated ovaries with RNase A. Indeed, after treatment with RNase A, HA-DIP1 nuclear foci became diffused compared to mock treated control ovaries (Supplementary Fig. 1s, t). These data are consistent with the idea that maintenance of DIP1 foci may require the transcripts that are present in the nucleus.

**Satellite bodies decorate the fourth chromosomes**. We characterized the DIP1 nuclear foci more carefully by examining the huge nurse cells of the later stage egg chambers, which allow better resolution of the nuclear localization. In stage 7/8 nurse cells, when the chromosomes become polyploid, DIP1 localizes as discrete round bodies around a DAPI dense region (Fig. 5a, *arrowhead*). This DAPI dense region was previously shown to be the heterochromatin of the fourth chromosomes and each

nurse cell contains only one such region because homologous chromosomes are paired and clustered during this stage of oogenesis[50]. Staining with a marker for the fourth chromosome Paint of fourth (POF) confirmed that this DAPI dense region is the chromosome four (Fig. 5b, *arrowhead*)[51]. HA-DIP1 also localized as bodies that decorated the fourth chromosomes (Fig. 5c, *arrowheads*). In general, the staining pattern of DIP1 differs from that of POF. While POF localizes on the chromosomes, DIP1 appears as small foci that surround or decorate the chromosomes. We named these DIP1 defined bodies satellite bodies because they resemble satellites that revolve around the planet (DAPI-dense dot).

Satellite bodies appear to define a novel class of nuclear bodies as other known nuclear bodies such as Cajal body and HLB do not associate with chromosome four (Fig. 5d, e, *arrowheads*)[44]. When the chromosomes become polytenized in the stage 10/11 nurse cells, satellite bodies become elongated in shape. Nevertheless, they still associate with the fourth chromosome as evident by the presence of POF staining around the DAPI-dense region (Fig. 5f–h, *arrowheads*).

Cytoplasmic bodies such as the nuage and balbiani bodies are only present in germline cells[48]. Besides germline cells, satellite bodies are also present in the somatic follicle cells. The fourth chromosomes are almost always associated with a DIP1-positive satellite body (Fig. 5i, *white arrowheads*). In some cells, there are also additional DIP1-positive bodies that are not associated with chromosome four (Fig. 5i, *yellow arrowheads*). The genomic association of these bodies is currently unclear and requires further investigation. This localization pattern is different from that of POF, which only labeled the fourth chromosomes (Fig. 5j, *arrowheads*). Similar bodies that associate and do not associate with the fourth chromosomes were also found in cultured S2 cells (Fig. 5k, l).

**DIP1 suppresses the abundance of INE-1 sisRNAs**. Since the localizations of the nucleolus and HLB correlate with a high concentration of transcription of specific RNA species, we asked whether the fourth chromosome could be a rich source of a specific class of RNA molecules. The *Drosophila* chromosome four is a very small chromosome and contains a very high density of the INE-1 transposable element sequences (Fig. 6a and Supplementary Fig. 5)[52–54]. It is also rich in essential genes and is transcriptionally active. Majority of the INE-1 elements are present in the introns of the genes, in either single or multiple copies, and are present in both orientations (Fig. 6b and Supplementary Fig. 5)[53]. Since intronic INE-1 sequences are co-transcribed with the host genes, chromosome four may be a rich source of INE-1 sisRNAs.

Indeed, INE-1 containing intronic transcripts are processed into sisRNAs in unfertilized eggs, which store a pool of stable and mature RNAs without any contamination from nascent transcripts. We detected the presence of INE-1 sisRNAs in unfertilized eggs via RT-PCR using generic primers for INE-1 sequences and intron-specific primers for two chromosome four genes *eyeless* (*ey*) and *CamKII* (Fig. 6b, c). The *CamKII* sisRNA was also previously detected in eggs to be ~10% the level of its parent gene[5]. Interestingly, both strands of the INE-1 sisRNAs from the *ey* gene were present in the eggs (Fig. 6d), implying that they may form dsRNAs. Fluorescent in situ hybridization showed that INE-1 sisRNAs clustered around and on the fourth chromosomes, although they were also present, to a lesser extent, in other regions of the nucleus (Fig. 6e). These INE-1 sisRNAs are closely associated with the DIP1-positive satellite bodies (some co-localize with DIP1, while others in close proximity to DIP1), suggesting some regulatory relationships (Fig. 6f). Furthermore,

we also detected interaction between DIP1 and INE-1 sisRNAs in S2 cells (Fig. 1c).

Since DIP1 represses *sisR-1*, we examined if DIP1 also regulates INE-1 sisRNA abundance. In *DIP1* mutant ovaries, the steady-state abundance of INE-1 sisRNAs was upregulated, suggesting that DIP1 represses INE-1 sisRNAs (Fig. 6g). We did not observed a significant change in the expression of *HeT-A* (a telomeric retrotransposon regulated by the piRNA pathway) (Fig. 6g)[48], suggesting that DIP1 is not a component of the piRNA pathway. We also did not observed any changes in the expression of two other chromosome four genes (*CG32000* and *CG2316*) that have INE-1 sequences in their introns (Fig. 6g), suggesting that DIP1 does not regulate the transcriptional state of the fourth chromosomes in general. Furthermore, overexpression of DIP1 in the germline cells led to a decrease in the level of INE-1 sisRNAs, but not for parental genes such as *ey* and *CamKII* (Fig. 6h), confirming a role of DIP1 in repressing INE-1 sisRNAs. Together, our data reveal the identification of DIP1 as a repressor of *sisR-1* and INE-1 sisRNAs at the post transcriptional level (Fig. 6i). Although DIP1 seems to regulate INE-1 sisRNAs in the satellite bodies, the regulation of *sisR-1*, which is not on the fourth chromosome, by DIP1 presumably occurs outside the satellite bodies.

## Discussion

Our results reveal the importance of the regulation of sisRNA activity/expression in GSC–niche occupancy. We propose that the *sisR-1* axis maintains GSCs in the niche, however, uncontrolled accumulation of *sisR-1* due to its unusual stability can lead to increase number of GSCs at the niche. DIP1 in turn limits the build-up of *sisR-1* to maintain ~2 GSCs per niche (Fig. 3s). GSC–niche occupancy is highly regulated by homeostatic mechanisms via negative feedback loops at the cellular and molecular levels[33, 55, 56]. Mis regulation of the niche may pose a problem as it allows for a greater chance of GSCs to accumulate mutations that may lead to tumor formation[31, 56, 57]. On the other hand, mechanisms that promote GSC–niche occupancy may be important to facilitate the replenishment of GSCs during aging[34, 58, 59]. Understanding the control of stem cell-niche occupancy will provide important insights to reproduction, cancer, and regenerative medicine.

In a large-scale RNAi screen for genes that regulate GSC self-renewal and differentiation, *rga* was identified as a gene required for GSC differentiation[29]. How *rga* regulates GSC self-renewal is currently unknown but our data suggest that GSC–niche adhesion and Mei-P26 are involved. The *rga* gene encodes for the NOT2 protein in the CCR4 deadenylase complex[60–62]. Surprisingly, studies have shown that other components of the CCR4 complex such as CCR4, Not1, and Not3 function in promoting GSC maintenance[35, 36]. Interestingly, Twin has been proposed to function with distinct partners to mediate different effects on GSC fates[35]. This suggests that other components such as CCR4 can also have additional functions outside the CCR4–NOT deadenylase complex in mediating GSC maintenance, thus affecting GSCs in opposite ways to Rga.

Here, we put forward a proposed model for *sisR-1*-mediated silencing. We hypothesize that folded *sisR-1* harboring a 3′ tail may form a ribonucleoprotein complex, which confers its stability, and allows scanning for its target via its 3′ tail. Binding of the 3′ tail to the target may promote local unwinding of *sisR-1* as the 3′ end of *ASTR* invades to form a more stable 76 nt duplex. We show that, in principle, it is possible to design a chimeric sisRNA to target a long ncRNA of interest such as *rox1*. In future, sisRNA can be potentially developed as tools to regulate nuclear RNAs of interest. Clearly, the efficiency and specificity of sisRNA-mediated

silencing need to be optimized. Because sisRNA-mediated target degradation requires a more extensive base-pairing between sisRNA and the target, the chances of off-target effects ought to be lower than siRNAs and antisense oligonucleotides. In broader terms, our study provides a paradigm, which encourages exploration of whether other sisRNAs or ncRNAs utilize a similar silencing strategy as *sisR-1*.

We describe a nuclear body (named satellite body) that associates with the fourth chromosomes. Satellite body adds to an existing group of nuclear bodies (nucleolus, HLB, and pearl) that associate with specific genomic loci[63]. It is generally believed that formation of such nuclear bodies correlates with a high concentration of RNA transcribed from the tandemly repeated gene loci. The formation of satellite bodies around the fourth chromosomes probably reflects a high concentration of DIP1 in regulating INE-1 sisRNAs transcribed there. The formation of satellite bodies may be promoted by the high concentration of INE-1 sisRNAs transcribed on the fourth chromosomes, and may facilitate the decay of INE-1 sisRNAs (Fig. 6i). We speculate that in the nucleoplasm, DIP1 that does not form observable satellite bodies is sufficient to regulate sisRNAs such as *sisR-1* transcribed from other chromosomes. Since DIP1 is a dsRNA binding protein, it may bind to mature sisRNAs to destabilize them. It may do so by recruiting RNA degradation factors (such as nuclear exosomes) or introducing RNA modification to "mark" sisRNAs for degradation. In future, it will be important to identify more components of the satellite bodies and their dynamics during differentiation and in response to stimuli in order to better understand the molecular mechanism of sisRNA metabolism.

## Methods

**Fly strains**. The following fly strains were used in this study: *y w, MTD-Gal4*[64], *da-Gal4, sisR-1 RNAi-1/CyO*[5], *sisR-1 RNAi-2/CyO*[5], *ASTR RNAi/TM3*[5], *UAS-sisR-1*[5], *rga RNAi GL00386* (Bloomington #35460), *DIP1^EY02625* (Bloomington #15577), *DIP1 RNAi GL00242* (Bloomington #35333), *dad-GFP* (enhancer trap), *P[bam]-GFP, mei-P26^mfs1* (Bloomington #25919), *UASp-shg* (Bloomington #58494), *dicer-2^L811fsX 16* and *ago1*[465]. Before dissection, females were fed with wet yeast for 7 days (7-day old) at 25 °C. Note that it has been verified previously that *sisR-1* RNAi knocked down *sisR-1* but not the parental *rga* mRNA[5]. Locations of RNAi constructs were indicated in Fig. 1c.

For generation of *sisR-1* overexpression clones, *hsFLP;nos>STOP>Gal4,UAS-GFP* flies[66] were crossed to *UAS-sisR-1* flies. Flies were heat shocked at 37 °C two times daily at 7–8 h intervals for three consecutive days and dissected for staining one day after the last heat shock.

In the overexpression assays, *da-Gal4* was used when combined with UASp transgenes, while *MTD-Gal4* was used when combined with UASt transgenes, to achieve maximum overexpression efficiencies.

**Generation of transgenic flies**. PCR of *rga* full-length coding sequence (CDS) was performed using primers, CACC-*rga* FW (5′ CACCATGGCGAATTTAAATTTTC AACAACCC 3′) and *rga* Rv (5′ TTATACAGACTGTCCATTCATAAACGCACT TATATTGG 3′). PCR of *ASTR* full-length sequence was performed using primers, CACC-*ASTR* FW (5′ CACCCAAAGTTAATCAGATATTCGGGTGG 3′) and *ASTR* Rv (5′ TGCAATCTCATTTACTTTGAAACATGAT 3′). For *UASp-HA-DIP1-c* transgenic flies primers used were cacc-DIP1-c Fw CACCATGAAGCGAA ATCGTCGTGC and DIP1-c Rv AGTGGTGTCGCTGTAGGTGA. Transgenic flies expressing DIP-1-c-HA were not generated because C-terminal tagged DIP1 protein was not stable when expressed in S2 cells. The PCR products were purified, cloned into pENTR TOPO vector (Invitrogen) and transformed into One Shot chemically competent *Escherichia coli* cells. Plasmids were checked by sequencing. LR reactions were then carried out using Gateway LR Clonase II Enzyme mix (Invitrogen). Transgenic flies were generated by BestGene using P-element-mediated insertion[67].

Transgenic flies expressing *dsRed-sisR-1-myc* was generated as previously described[5]. Intronic fragments were chemically synthesized and sub-cloned into linearized UASt-dsRed-myc plasmid. Injection was done by BestGene Inc.

**DIP1 antibody**. Two independent affinity-purified polyclonal antibodies against full-length DIP1-c were generated in rabbits by GenScript. Both antibodies gave identical staining patterns.

**Immunostaining**. Immunostaining was performed as described previously[68]. Ovaries were fixed in a solution of 16% paraformaldehyde and Grace's medium at a ratio of 2:1 for 10–20 min, rinsed and washed with PBX solution (phosphate-buffered saline containing 0.2% Triton X-100) three times for 10 min each, and pre-absorbed for 30 min in PBX containing 5% normal goat serum. Ovaries were incubated overnight with primary antibodies at room temperature, washed three times for 20 min each with PBX before a 4 h incubation with secondary antibodies at room temperature. Ovaries were again washed three times for 20 min each with PBX. Primary antibodies used in this study are as follows: mouse monoclonal anti-α-Spectrin (3A9, 1:1; Developmental Studies Hybridoma Bank), rabbit anti-Rga (1:500, kind gift from Claudia Temme and Elmar Wahle)[61], guinea pig anti-Vasa (1:1000, kind gift from Toshie Kai), rabbit anti-pMad (1:50; Cell Signaling Technology #9516), mouse anti-Bam (1:10, Developmental Studies Hybridoma Bank), mouse anti-GFP (3E6, 1:500, Invitrogen #A11120), rat anti-E-Cadherin (1:200, Developmental Studies Hybridoma Bank), rabbit anti-Mei-P26 (1:1000, kind gift from Paul Lasko)[38], rabbit anti-DIP1 (1:100, this study), rabbit anti-Coilin (1:2,000, kind gift from Joseph Gall), rabbit anti-LSM10 (1:2,000, kind gift from Joseph Gall), rabbit anti-POF (1:1,000, kind gift from Jan Larsson) and rat anti-HA (3F10, 1:500, Sigma #11867423001). Labeling of apoptotic cells were done using the TUNEL assay (Molecular Probes) in accordance to the manufacturer's protocol. Images were taken with a Carl Zeiss LSM 5 Exciter Upright microscope and processed using Adobe Photoshop.

**RNA FISH**. RNA FISH was performed according to a published protocol[69]. DIG-labeled DNA probes for INE-1 were generated by PCR as described previously[5]. Ovaries were dissected in Grace's medium, fixed for 20 min in PBTT with 4% formaldehyde. Ovaries were then rinsed and washed with PBT and followed by 10 min incubation in cold 80% acetone at −20 °C. The ovaries were washed and post-fixed for 10 min in PBT with 4% formaldehyde, followed by washing with PBT, rinsing in 1:1 PBT/hybridization solution, hybridization solution, and blocking in hybridization solution for at least 2 h at 42 °C. The samples were then incubated in the probe/hybridization mixture at 42 °C overnight. Next day, samples were washed with 3:1, 1:1, 1:3 hybridization solution/PBT and PBT at 42 °C, and then blocked and incubated in primary antibodies (anti-DIP-POD and rabbit anti-DIP1) for 2 h at room temperature. The samples were washed and Tyramide Signal Amplification was performed in accordance to the manufacturer's protocol.

**Actinomycin D treatment**. Actinomycin D treatment was performed as previously described[5]. Ovaries were incubated in Grace's medium containing 20 μg ml⁻¹ actinomycin D with constant rocking at room temperature.

**RNase A treatment of ovaries**. Ovaries were dissected in Grace's media at room temperature. Ovaries were incubated in Grace's media containing either no chemicals (mock) or RNase A (100 μg ml⁻¹) with rocking for 30 min at room temperature as described previously[5].

**RNA extraction**. Tissues were homogenized in 1.5 ml Eppendorf tubes using a plastic pestle and RNA was extracted using the TRIzol extraction protocol (Ambion) or the Direct-zol RNA miniprep kit (Zymo Research). RNA was quantified using a Nanodrop spectrophotometer to ensure equivalent loading for subsequent experiments.

**RNA immunoprecipitation**. S2 cells that survive in serum-free medium were obtained from Steve Cohen's laboratory. Immunoprecipitation was performed as described with minor modifications[70]. Cells were lysed in protein extraction buffer (50 mM Tris-HCl pH 7.5, 150 mM NaCl, 5 mM MgCl$_2$, 0.1% NP-40) supplemented with Protease Inhibitor Cocktail (Roche). Lysates were then pre-cleared with protein A/G agarose beads (Merck Millipore). Three microlitre of rabbit anti-DIP1 was added and incubated for 3 h at 4 °C. Protein A/G agarose beads were then added and incubated for another 1 h. After incubation, beads were washed three times with protein extraction buffer, and RNA was extracted using the Direct-zol RNA miniprep kit (Zymo Research).

**RT-PCR**. RT-PCR was performed as previously described[5]. For standard RT-PCR, total RNA was reverse transcribed with random hexamers for 1 h using AMV-RT (New England Biolabs), M-MLV RT (Promega) or Superscript III (Invitrogen). Strand-specific RT-PCR was performed using SuperScript III RT (Invitrogen) or M-MLV RT (Promega). PCR was carried out using the resulting cDNA. For quantitative PCR (qPCR), SYBR Fast qPCR kit master mix (2×) universal (Kapa Biosystems, USA) was used with addition of ROX reference dye high and carried out on the Applied Biosystems 7900HT Fast Real-Time PCR system. Oligo sequences were reported previously[5]. *DIP1* Fw: 5′ TAATACGACTCACTATAG GGAGAAAGAAGTTGCGACAGAACCG 3′ and *DIP1* Rv: 5′ TAATACGACT CACTATAGGGGAGACGAACAGCTTGTAGATGGCA 3′. CamKII INE-1 Fw TGGGCTATTTTTAGGCGTCA, CamKII INE-1 Rv TATGAACGCGTCGATC TCAG, ey INE-1 Fw CGGAAAATGCCAAGGACTAA, ey INE-1 Rv GCTAAA TGGGCACACTCGTC, INE-1 Fw GGCCATGTCCGTCTGTCC, INE-1 Rv AGCTAGTGTGAATGCGAACG, *rox1* forward TGCAGTGGCAGTTTCTTCTG,

*rox1* reverse GGTCCGTGCAAAGCAG TAAT, *rox2* forward TCTCCGAAGCAAAATCAAGC, *rox2* reverse TGTTGCGTTCCAAGACACAT.

**Western blot**. Ovaries were dissected in Grace's medium and homogenized in 2× sample buffer. Western blotting was performed as previously described[68]. Antibodies used were rabbit anti-Rga[61] (1:5000), mouse anti-FLAG M2 (1:5000, Sigma #F3165), mouse anti-Alpha Tubulin (clone DM1A, 1:10,000, Millipore #05-829), and rat anti-HA (3F10, 1:5,000, Sigma #11867423001). All uncropped western blots can be found in Supplementary Fig. 6.

**Northern blotting**. Northern blotting was performed as described previously[5]. DIG-labeled DNA probes were made by PCR using genomic DNA as the template. To detect *sisR-1*, RNA was run on an 8% polyacrylamide gel (8 M urea, 1× TBE buffer), and transferred onto a nylon membrane by electrophoresis. To detect, *ASTR*, RNA was run on a 0.8% agarose/formaldehyde gel and transferred onto a nylon membrane by capillary action. RNA was UV crosslinked to the membrane, pre-hybridized in salmon sperm DNA, and hybridized with probes in DIG Easy Hyb Granules (Roche) at 42 °C overnight. Next day, the membranes were washed once with 2× SSC and 0.1% SDS, and twice with 0.1× SSC and 0.1% SDS, followed by detection with the CDP-Star chemiluminescent substrate (Roche).

**Bioinformatics**. Annotation information of INE-1 all insertions and gene introns are from FlyBase. Bedtools intersect was used to extract INE-1 insertions located at introns. Custom scripts were used to convert the raw annotation data to bed format to upload into UCSC GB as custom tracks. The images were downloaded from custom tracks from UCSC GB. The custom tracks represent the INE-1 all insertions in the dm3 genome or insertions only in annotated introns.

**Statistical analysis**. At least 20 germaria from a total of 3–4 flies (6–8 ovaries) were sampled for each genotype. Fisher's exact test was performed for all GSC counts except for Fig. 3n where Wilcoxon test was performed. No randomization or blinding was performed.

**Data availability**. The authors declare that all data supporting the findings of this study are available within the article and its supplementary information files or from the corresponding author upon reasonable request.

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

## Acknowledgements

We are grateful to Yu Cai, Richard Carthew, Joseph Gall, Toshie Kai, Jan Larsson, Paul Lasko, Katsutomo Okamura, Haruhiko Siomi, Mikiko Siomi, Claudia Temme, Elmar Wahle, and Yukiko Yamashita for sharing reagents, antibodies and flies, Li Zhou and Ryan Teo (Okamura laboratory) for assistance in INE-1 bioinformatics analysis, Siqi Ng for help on RT-PCR, and Ismail Osman for comments on the manuscript, the Developmental Studies Hybridoma Bank (DSHB) and the Bloomington Stock Center. We acknowledge the microscopy facility at the Temasek Life Sciences Laboratory for their support over the course of this study. The authors are supported by the Temasek Life Sciences Laboratory.

## Author contributions

J.T.W. and F.A. contributed equally to this work. J.T.W. performed *sisR-1* genetic analyses. F.A. performed DIP1 genetic and immunofluorescence analyses. A.Y.E.N. performed northern blotting, generated DIP1-overexpression transgene and contributed to DIP1 genetic studies. M.L.-I.T. generated Rga-overexpression transgene and performed some control qPCR. G.J.E.L. contributed to DIP1 genetic studies. J.W.P. conceived the study, performed remaining experiments, analyzed the results with the authors, and wrote the paper with inputs from the authors.

## Additional information

**Competing interests:** The authors declare no competing financial interests.

