## [Peer Review file · Nature Communications]

Reviewers' comments:

Reviewer #1 (Remarks to the Author):

Recent studies have shown that a large portion of our genome is transcribed to produce a number of long non-coding RNAs. However, the functionality of only a small number of these long non-coding RNAs has been demonstrated.

This manuscript continues the Pek lab's analysis of stable intronic sequence RNAs (sisRNAs). The authors here identify a dsRNA-binding protein DIP1 that negatively regulates the abundance of sisRNAs. This regulation may be important to maintain female germline stem cells homeostasis.

Comments:

1. Figure 1: It is unclear how the authors identified DIP1 as a candidate gene that functions to repress the expression of sisRNAs. The authors should describe in detail how they identified DIP1. There are a number of dsRNA-binding proteins in *Drosophila* (Lasko P, JCB 2000). Is DIP1 only a dsRNA-binding protein, among them, that affects the abundance of sisRNAs?

2. Figure 2: It is unclear how the formation of "satellite body" is related to the function of DIP1 or sisRNAs? Are the DIP1 aggregates formed in S2 cells in which DIP1 interacts with one of sisRNAs? Are targets of sisRNAs such as ASTR accumulated in the aggregates? What about rga pre-mRNA, a target of ASTR? Where in the nucleus are they localized?

3. Figure 3: It is unclear how the finding of INE-1-containing sisRNAs is related to the rest of the story. What might be the function of these INE-1 sisRNAs? It appears that INE-1 sisRNAs are accumulated in or near satellite bodies (Figure 3 F). But these results present non-quantified data from a single experiment. sisR-1 appear to be derived from the third chromosome but not from the fourth chromosome. Is sisR-1 accumulated in satellite bodies? Do all sisRNAs behave similarly? Do sisRNAs have something in common except their origins being introns. Overexpression of DIP1 leads to a decrease on INE-1 sisRNAs. Does it also affect the abundance of their source genes, *eyeless* and *CamKII*, for example? Does overexpression of other nuclear dsRNA-binding proteins affect the abundance of sisRNAs and their targets?

4. Figure 4 & 5: These figures show the results of knock down or overexpression of sisR-1. But we aren't even assured that sisR-1 was knocked down or overexpressed. Are overexpressed sisR-1 and its mutant forms located in satellite bodies? Why and how is the 3' end of sisR-1 is required for its function? How could sisR-1, together with DIP1, regulate the expression of ASTR?

5. No Figure 4 R in Figure 4 (p10, line 199).

In sum, results shown in the manuscript are not well connected with each other and are preliminary.

Reviewer #2 (Remarks to the Author):

This manuscript presents a notably complete story (prompted and buttressed by some earlier findings) concerning a modifier (DIP1) of the levels of a stable intronic sequence RNA (sisR-1) and the effects of DIP1 and sisRNA-1 levels on effectors (rga gene product via an intermediate RNA regulator, ASTR) that have a big effect on female germline stem cell (GSC) maintenance (largely via changes in the level of Cadherin and Mei-P26). The supporting evidence mostly appears strong to me, so I think that this work provides a robust, quotable example regarding the functional impact of sisRNAs. My subjective response was surprise at such a strong biological effect, although I can see (retrospectively) that evidence for a large regulatory effect of sisR-1 on Rga, and of the

CCR4 complex (which can contain Rga) on GSC biology has been presented elsewhere and therefore set the stage for the reported outcomes. The localization of DIP1 to nuclear regions around the 4th chromosome in germline and other cells is also reported and of some interest (for the future). I think this information is fine to include even though it interrupts the sisR-1 story.

Most of my comments concern clarifications, including a frequent need (maybe in more places than I have cited) for fuller introductions to what was previously established.

56. Explain the screen.

57 "repress... expression" implicates transcription for most readers- "reduced the levels" may be more accurate description of the results

59 Is there any in situ information on sisR-1 changes? If RNA levels were too low to visualize I think that is worth stating somewhere.

70 It seems odd that sisR-1 levels actually increase with actD. Please confirm that this is not an artifact of normalization and provide some possible explanation or precedent. Was the experiment also done for DIP1 loss of function?

In Fig. 1A, B there should be a numerical measure of band intensities (& information about any repetitions), the 5S rRNA control is obviously not easily quantified by eye or other means, and was RT-PCR to sisR-1 an alternative, more quantitative, way to measure (without rga RNA interference)?

In Fig. S1Q,R there may be an error in labeling, text or legend because there does not seem to be a direct comparison of HA-DIP1 (by HA) and total DIP1 protein, as promised by "the same nuclear foci". In this case, I would say it is genuinely difficult to know if the HA staining and DIP1 staining in different images have the same patterns.

117 I think it is worthwhile for the authors to explain a little more carefully the disposition of DIP1 staining relative to DAPI and POF to define it as (perhaps) surrounding DNA and chromatin rather than overlapping, or whatever they find to be most accurate for christening this localization pattern.

143 Previously, sisRNAs were carefully described according to numerous criteria. Do the INE-1 sisRNAs meet these criteria or are they just detectable by RT-PCR and in situ? Also, in Fig. 3D "gDNA" is not explained in the legend.

250 INE sisRNA "closely associated" with DIP1. Do they co-localize with POF, DAPI to allow a fuller explanation of the relative location of chromatin, RNAs and DIP1 protein?

177 I think it is worth explaining prior evidence that RNAi targeting of sisRNAs, or this one in particular, is selective, without generally affecting the parent protein-coding transcript (at least in a simple, direct negative way).

189 There is some increase in GSC # with the altered sisR-1 (modify "no increase")

202 Was it also observed here (as well as in the cited paper) that overexpression of Shg did not increase GSC #? Was the magnitude of increased DE-cadherin caused by Shg and sisR-1 compared (presumably the former is higher)?

217 Explain prior ASTR evidence as an intermediate between sisR-1 and rga better here or earlier.

219 Explain "early & late stage" ovaries of S4C.

224 Although ASTR RNAi greatly modifies the effect of sisR-1 RNAi, the converse is also true to some (lesser) degree. This merits a few words about whether this is because null conditions are not achieved or sisR-1 might have additional targets.

228 I cannot find in ref 36 a description of the rga/Not2 mutant phenotype. If it is there please cite in a way it might be found but (as will be noted later) the majority of that paper is about other potential partners which have an opposite phenotype. Probably best to omit the reference at this point and discuss the whole thing later.

229 It is surprising to me that Rga overexpression is less effective than sisR-1 RNAi at reducing GSC #, raising the question of how much Rga protein changes in each case. Really, in situ staining

would be needed to answer that question for GSCs but looking at Fig. S4F, G ought to give a clue. However, Rga seems reduced by sisR-1 RNAi! Surely, that is contrary to all other evidence?

267 It is mentioned that rga is not on the 4th but in a very subtle way. I think it should be made clearer to readers that the satellite body data are quite distinct from the rest of the functional data. Because of this I find whatever is to be concluded about satellite bodies to be relatively weak. The authors can certainly speculate that there may be similar "bodies" in other locations that are harder to see (and affect rga) but it must be clear that it is speculation.

I find the most puzzling aspect of the paper to be how results relate to prior studies of CCR4/Not components. I think the sum of past papers could be explained a little more carefully because it is not uniform and contains plenty of information relevant for Shg, Mei-26 mechanisms. Indeed, it does not seem right to just conclude Rga acts independently of other complex members in the light of evidence that those mutations also affect GSCs via DE-cadherin and Mei-26, albeit in an opposite manner.

Another open question meriting at least speculation is how DIP1 reduces sisR-1 levels.

The statistics for graphs like Fig. 4Q, 5J... do not seem to be reported appropriately. More words are needed, exact n values should be given and it seems that the asterisks apply always to the number of occurrences of a single number of GSCs, where Fisher's exact test rather than a t test seems appropriate.

In summary, I believe that the specific comments above can readily be addressed, as they basically ask for better explanations and a little more extant information. Besides that, I find the evidence supports a story of suitable interest to a wide audience. I will be interested to see if other reviewers have critical comments regarding some of the RNA methodology or CCR4 biology that reflect a deeper expertise.

General response: We would like to express our gratitude to the reviewers for their time to provide constructive suggestions, and the opportunity to revise the manuscript to address their concerns. We have now addressed all the comments (see point-by-point below) by providing more experimental data, analyses and textual clarifications. We hope that the reviewers find our response satisfactory.

Reviewer #1 (Remarks to the Author):

Recent studies have shown that a large portion of our genome is transcribed to produce a number of long non-coding RNAs. However, the functionality of only a small number of these long non-coding RNAs has been demonstrated.

This manuscript continues the Pek lab's analysis of stable intronic sequence RNAs (sisRNAs). The authors here identify a dsRNA-binding protein DIP1 that negatively regulates the abundance of sisRNAs. This regulation may be important to maintain female germline stem cells homeostasis.

Comments:

1. Figure1: It is unclear how the authors identified DIP1 as a candidate gene that functions to repress the expression of sisRNAs. The authors should describe in detail how they identified DIP1. There are a number of dsRNA-binding proteins in *Drosophila* (Lasko P, JCB 2000). Is DIP1 only a dsRNA-binding protein, among them, that affects the abundance of sisRNAs?

Response: We began by hoping to find candidate genes that are specific to the sisRNA pathway. We referred to the Lasko, JCB 2000 paper and found 7 dsRNA binding proteins that localize to the nucleus (by cross-checking with FlyBase and published literature). Over the years, many of the original dsRNA binding proteins have been assigned functions to miRNA and RNAi pathways. For example, CG188 (now *pasha*) and *drosha*, and CG12493 (now *blanks*) have been assigned to the miRNA and siRNA pathways respectively. CG12598 (now *adar*) and *mle* have been shown to be required for RNA editing and X-activation respectively. Among the 7, DIP1 and CG8273 are the least characterized and therefore most promising. By performing northern blotting, we found that DIP1 and CG8273 are both negative regulators of sisR-1 abundance. In this paper, we focus on DIP1, and will report the characterization of CG8273 in another study. Our approach does not rule out the possibility that the other dsRNA binding proteins also regulate sisRNAs. Because they already have well-established molecular roles, it is difficult to address if their impact on sisRNAs, if any, is specific or direct. One example is *Adar* that has been shown to regulate heterochromatin status of the fourth chromosomes¹. We have described our candidate selection in greater details as follows:

(page 4, line 3) We began by searching for candidate genes that are specific to the sisRNA pathway. In *Drosophila*, there are seven dsRNA binding proteins that localize to the nucleus. Over the years, many of the original dsRNA binding proteins have been assigned functions to the miRNA and RNAi pathways. For example, CG188 (now *pasha*) and *drosha*, and CG12493 (now *blanks*) have been assigned to the miRNA and siRNA pathways respectively. CG12598 (now *adar*) and *mle* have been shown to be required for RNA editing and X-activation respectively. Among the seven, DIP1 and CG8273 are the least characterized and therefore most promising.

2. Figure 2: It is unclear how the formation of “satellite body” is related to the function of DIP1 or sisRNAs? Are the DIP1 aggregates formed in S2 cells in which DIP1 interacts with one of sisRNAs? Are targets of sisRNAs such as ASTR accumulated in the aggregates? What about rga pre-mRNA, a target of ASTR? Where in the nucleus are they localized?

Response: We deeply apologize for the confusion regarding the interpretation on “satellite body” (related to original Figures 2 and 3). The reviewer appears to think that we are proposing satellite bodies as sites of sisRNA-mediated silencing of ASTR, therefore asking for the localizations of sisR-1 and ASTR in the bodies. We clarify that we do *not* intend to propose that satellite bodies as sites of sisRNA-mediated ASTR silencing. This is supported by the fact that we were not able to detect ASTR in DIP1 immunoprecipitates (new data in Figure 1c), which is not consistent with the idea that DIP1-sisR-1 complex mediate ASTR silencing. We describe the new results as follows:

(page 5, line 14) Moreover, DIP1 did not interact with *ASTR*, a target of *sisR-1*, suggesting that DIP1 is not part of the *sisR-1* silencing complex (Fig. 1c).

We clarify that in this manuscript, we report the correlation of satellite body formation at regions of concentrated INE-1 sisRNA production. Our most parsimonious explanation is that DIP1 accumulates at higher concentration at sites of concentrated INE-1 sisRNAs (from Chromosome 4), and possibly degrades INE-1 sisRNA there. Our hypothesis is based on the following observations:

1. Chromosome 4 is very small and has many intronic INE-1 sequences (Figure 6a; Supplementary Figure 5).
2. INE-1 RNAs are present at high concentration around Chromosome 4 (Figure 6e, f).
3. DIP1 is concentrated around Chromosome 4 (Figures 5, 6f) (Note that diffused DIP1 staining is also present in the nucleoplasm at low levels).
4. Formation of DIP1 satellite bodies depends on RNA (Supplementary Figure 1s, t).
5. DIP1 binds to INE-1 sisRNA (Figure 1c).

Our data suggest that satellite bodies reflect concentrated areas of high sisRNA degradation, and in the *Drosophila melanogaster*, it appears to be the small Chromosome 4 where INE-1 sisRNAs are actively being produced. We also hypothesize that nucleoplasmic DIP1 that does not form microscopically observable satellite bodies, is sufficient for degrading sisRNAs from other chromosomes such as sisR-1 from Chromosome 3. We have discussed more carefully as follows:

(page 18, line 10) We describe a nuclear body (named satellite body) that associates with the fourth chromosomes. Satellite body adds to an existing group of nuclear bodies (nucleolus, HLB and pearl) that associate with specific genomic loci. It is generally believed that formation of such nuclear bodies correlates with a high concentration of RNA transcribed from the tandemly repeated gene loci. The formation of satellite bodies around the fourth chromosomes probably reflects a high concentration of DIP1 in regulating INE-1 sisRNAs transcribed there. The formation of satellite bodies may be promoted by the high concentration of INE-1 sisRNAs transcribed on the fourth chromosomes, and may facilitate the decay of INE-1 sisRNAs (Fig. 6i). We speculate that in the nucleoplasm, DIP1 that does not form observable satellite bodies is sufficient to regulate sisRNAs such as sisR-1 transcribed from other chromosomes.

Perhaps it is not surprising to find that the formation of some nuclear bodies are not necessary for their functions. An excellent example is the Cajal body, which contains high concentrations of snRNAs, snoRNAs and their associated proteins required for RNA modification. However, it was found that in coilin mutants, where the assembly of Cajal bodies are abolished, snRNA modification remains unaffected². Nevertheless, future studies will be required to investigate whether satellite body formation is necessary for INE-1 sisRNA degradation. Molecular mutagenesis analyses of DIP1 to link INE-1 sisRNA binding, satellite body formation, and activity will be required. Such investigations seem to be beyond the scope of the current paper since we do not claim that satellite body formation is necessary for DIP1 function.

Although not the focus of this paper, we agree with the reviewer that localizations of sisR-1, ASTR and rga pre-mRNA in the cells by FISH are interesting data. However, also highlighted by Reviewer #2, FISH on these low abundance RNA molecules are currently technically challenging on our hands. We have not been able to obtain specific signals for sisR-1 above background using conventional FISH. We would like to carry out single molecule FISH and report the findings as a separate future study.

We realize that the flow of the manuscript may have misled the Reviewer to think that satellite bodies are sites of sisR-1-mediated ASTR silencing. We have moved Figures 2 and 3 (satellite bodies and INE-1 sisRNAs) to the back. The new flow would allow us to establish our main story more clearly that (1) DIP1 regulates GSC via sisR-1, followed by (2) DIP1 as a general sisRNA regulatory protein by repressing INE-1 sisRNA. We have also deleted some general statements in the Discussion that can contribute to the misleading impression.

3. Figure 3: It is unclear how the finding of INE-1-containing sisRNAs is related to the rest of the story. What might be the function of these INE-1 sisRNAs? It appears that INE-1 sisRNAs are accumulated in or near satellite bodies (Figure 3 F). But these results present non-quantified data from a single experiment. sisR-1 appear to be derived from the third chromosome but not from the fourth chromosome. Is sisR-1 accumulated in satellite bodies? Do all sisRNAs behave similarly? Do sisRNAs have something in common except their origins being introns. Overexpression of DIP1 leads to a decrease on INE-1 sisRNAs. Does it also affect the abundance of their source genes, eyeless and CamKII, for example? Does overexpression of other nuclear dsRNA-binding proteins affect the abundance of sisRNAs and their targets?

Response: The concern about sisR-1 in satellite body is addressed above in our response to point #2. At present, we have no evidence to suggest that satellite body is a specialized body where all sisRNAs reside.

It is extremely interesting to know the function of INE-1 sisRNAs. However, we prefer to address this question in the future as the focus of this manuscript is on sisR-1. To address the reviewer's question, we have discussed the possible function of INE-1 sisRNAs as follows:

(page 19, line 9) During the course of characterizing DIP1, we unexpectedly discovered a class of INE-1 sisRNAs transcribed from the fourth chromosomes. Currently, we do not know the function of INE-1 sisRNAs. Following a recent study that suggested that certain sisRNAs can act to promote parental gene transcription during embryonic development, it is tempting to

speculate that INE-1 sisRNAs may function in a similar manner to modulate the transcriptional state of the fourth chromosomes during early development.

The observation of high concentration of INE-1 sisRNAs around the 4th chromosome is not surprising due to the high density of INE-1 sequences there. Furthermore, the visualization of INE-1 of around the 4th is consistent in different experiments (Figure 6e, f).

The reviewer raised a very interesting question regarding the definition and a common behaviour of sisRNAs. Since the coining of the term sisRNAs in *Xenopus tropicalis* by Dr. Joseph Gall in 2012, sisRNAs have been defined in quite broad terms as the name suggests: (1) stable for hours, (2) derived from introns³. Although different variants of sisRNAs have been identified, such as linear and circular forms, which can localize to the nucleus or cytoplasm, sisRNAs have not been reported to have a common molecular function, structure or cellular localization. Our data may provide first insights to a clearer definition of sisRNAs. We show that DIP1 binds to both sisR-1 and INE-1 sisRNAs, suggesting that some sisRNAs may be regulated by a common protein. In future, it will be very insightful to perform genome-wide deep sequencing of RNA bound by DIP1 to get a bigger picture, which is beyond the scope of the current manuscript.

Following the reviewer's suggestion, we have checked *eyeless* and *CamKII* expression in DIP1 overexpression ovaries. Unlike INE-1 sisRNAs, both genes are not downregulated, indicating specificity (Figure 6h). We have amended the text as follows:

(page 16, line 11) Furthermore, overexpression of DIP1 in the germline cells, led to a decrease in INE-1 sisRNAs levels, but not parental genes such as *ey* and *CamKII* (Fig. 5h), confirming a role of DIP1 in repressing INE-1 sisRNAs.

As discussed in our response to point #1, we do not exclude the possibility that other nuclear dsRNA binding proteins may also regulate sisRNAs, albeit in an indirect way. For example, Adar has been shown to globally promote chromosome 4 transcription by repressing the RNAi pathway¹. Thus, any impact on INE-1 sisRNAs by Adar overexpression is very likely to be indirect or difficult to interpret. In summary, our DIP1 mutant and overexpression analyses have resulted in consistent and specific effect on INE-1 sisRNAs but not their parental genes, therefore excluding non-specific effects. We have discussed this point as follows:

(page 19, line 3) It is useful to note that we do not exclude the possibility that other nuclear dsRNA binding proteins may also regulate sisRNAs, albeit in an indirect way. For example, Adar has been shown to promote global chromosome four transcription by repressing the RNAi pathway¹. Therefore, it is not surprising if Adar regulates INE-1 sisRNAs via the transcription of parental genes on chromosome four. Importantly, our data show that DIP1 negatively regulates INE-1 sisRNAs specifically with no effect on the parental gene expression.

4. Figure 4 & 5: These figures show the results of knock down or overexpression of sisR-1. But we aren't even assured that sisR-1 was knocked down or overexpressed. Are overexpressed sisR-1 and its mutant forms located in satellite bodies? Why and how is the 3' end of sisR-1 is required for its function? How could sisR-1, together with DIP1, regulate the expression of ASTR?

Response: The concern about *sisR-1* in satellite body is addressed above in our response to point #2. The *sisR-1* knockdown and overexpression flies had been reported and verified to work in an earlier study⁴. Moreover, note that the knockdown of *sisR-1* is specific to the *sisRNA* and not the pre-mRNA⁴. To make it clearer, we have included two statements as follows:

(page 6, line 5) We found that manipulating the expression of *sisR-1* using previously reported RNAi lines, which specifically knocked down *sisR-1* but not its parental gene, changed the GSC-niche occupancy.

(page 6, line 15) Transgene expression was verified by measuring *dsRed* expression (Supplementary Fig. 2c).

The reviewer raised an interesting question about the function of *sisR-1* 3' end. We initially thought that the question may be out of scope of the current manuscript, but we do appreciate the reviewer's suggestion to incorporate such data. Over the past 3 years, we had gathered and assembled some relevant data for another manuscript. Following the reviewer's suggestion, we now include those in this manuscript. We provide some genetic evidence to propose a model that *sisR-1* recognizes its target *ASTR* via its 3' tail, followed by formation of a more extensive double-stranded RNA intermediate, which is degraded by a double-stranded RNA endonuclease *Dicer-2*. The new data are now in new Figure 4 and Supplementary Figure 4, and explained in the text (Results and Discussion). In total, we show that:

1. The 3' tail of *sisR-1* is required for *ASTR* repression (Supplementary Figure 2a-c).
2. The 3' tail of *sisR-1* is not sufficient for target repression (Supplementary Figure 4).
3. A 76 nt 3' end of *sisR-1*, which overlaps with the target is sufficient for repression (Figure 4).

While we provide some evidence to suggest how the 3' end of *sisR-1* regulates *ASTR*, we acknowledge that further molecular and biochemical studies are needed to test the model. However, we believe that such studies are out of scope of this current paper. We hope that this new piece of data will provide insights to future elucidation of the mechanism using methods in molecular biology and biochemistry.

5. No Figure 4 R in Figure 4 (p10, line 199).

Response: Amended.

In sum, results shown in the manuscript are not well connected with each other and are preliminary.

Response: Thank you for pointing out that the original flow of the manuscript is confusing, and the opportunity to make it clearer. We have moved the original Figures 2 and 3 to the back, and inserted a new Figure 4 on the model for *sisR-1* silencing. Now the manuscript flows in this way: (1) *DIP1* regulates GSCs via *sisR-1*, (2) mechanism of *sisR-1*-mediated *ASTR* silencing, (3) *DIP1* also regulates *INE-1* *sisRNAs*.

Reviewer #2 (Remarks to the Author):

This manuscript presents a notably complete story (prompted and buttressed by some earlier findings)

concerning a modifier (DIP1) of the levels of a stable intronic sequence RNA (sisR-1) and the effects of DIP1 and sisRNA-1 levels on effectors (*rga* gene product via an intermediate RNA regulator, ASTR) that have a big effect on female germline stem cell (GSC) maintenance (largely via changes in the level of Cadherin and Mei-P26). The supporting evidence mostly appears strong to me, so I think that this work provides a robust, quotable example regarding the functional impact of sisRNAs. My subjective response was surprise at such a strong biological effect, although I can see (retrospectively) that evidence for a large regulatory effect of sisR-1 on *Rga*, and of the CCR4 complex (which can contain *Rga*) on GSC biology has been presented elsewhere and therefore set the stage for the reported outcomes. The localization of DIP1 to nuclear regions around the 4th chromosome in germline and other cells is also reported and of some interest (for the future). I think this information is fine to include even though it interrupts the sisR-1 story.

Response: Thank you for the very encouraging words! As discussed above, we have now moved Figures 2 and 3 to the back to avoid interruption to the sisR-1 story.

Most of my comments concern clarifications, including a frequent need (maybe in more places than I have cited) for fuller introductions to what was previously established.

56. Explain the screen.

Response: Explained. Please refer to our response to point #1 from reviewer #1.

57 “repress... expression” implicates transcription for most readers- “reduced the levels” may be more accurate description of the results

Response: Amended.

59 Is there any in situ information on sisR-1 changes? If RNA levels were too low to visualize I think that is worth stating somewhere.

Response: The reviewer is correct that the sisR-1 level is too low to achieve meaningful signals by fluorescent in situ hybridization (FISH). Furthermore, the results may be difficult to interpret as we cannot distinguish between sisR-1 and *rga* pre-mRNA by non-quantitative FISH. Following the reviewer’s suggestion, we have now state it as follows:

(page 4, line 18) However, we were not able to perform in situ hybridization to detect *sisR-1* in the ovaries due to its low abundance and inability to distinguish *sisR-1* from *rga* pre-mRNA.

70 It seems odd that sisR-1 levels actually increase with actD. Please confirm that this is not an artifact of normalization and provide some possible explanation or precedent. Was the experiment also done for DIP1 loss of function?

Response: We were also surprised by this unexpected result in the beginning. This effect was not an artifact of normalization and it is reproduced in biological replicates. The increase in sisR-1 levels after actinomycin D treatment is probably due to newly processed mature forms from the precursors present

at the start of the treatment. Consistent with this idea, we observed a concomitant decrease in *rga* pre-mRNA levels over the same time interval (Reviewer figure 1). Upon DIP1 overexpression, while more mature *sisR-1* are produced, at the same time, they also predominantly enter the degradation pathway, leading to a no change in steady-state levels (Reviewer figure 2).

Reviewer figure 1. RT-qPCR showing relative expression of *rga* pre-mRNA (normalized to *actin5C*) after actinomycin D treatment.

Reviewer figure 2. Overview of the possible fates of *sisR-1*.

Interestingly, a study reported in 1994 more than 20 years ago, documented that actinomycin D could inhibit the turnover of a nuclear form of *hsr-omega* noncoding RNA in *Drosophila*⁵. However, we are not aware of the molecular mechanisms of such an effect of actinomycin D on nuclear RNA turnover. We had only done the experiments for DIP1 overexpression ovaries as we think the readout will be more sensitive than DIP1 mutants. We have explained the observations in greater details as follows:

(page 5, line 2) In control ovaries, we observed a strong accumulation of *sisR-1* over time from 60 min to 120 min post treatment (Fig. 1b), indicating that mature and stable *sisR-1* was being produced from the precursors during this period of time.

In Fig. 1A, B there should be a numerical measure of band intensities (& information about any repetitions), the 5S rRNA control is obviously not easily quantified by eye or other means, and was RT-PCR to sisR-1 an alternative, more quantitative, way to measure (without rga RNA interference)?

Response: We have now used ImageJ to quantify the band intensities and represented the values below the blots. Information about the number of biological replicates was also added to the legends. Unfortunately, RT-PCR is not suitable to measure sisR-1 levels because we cannot distinguish it from rga pre-mRNAs.

In Fig. S1Q,R there may be an error in labeling, text or legend because there does not seem to be a direct comparison of HA-DIP1 (by HA) and total DIP1 protein, as promised by “the same nuclear foci”. In this case, I would say it is genuinely difficult to know if the HA staining and DIP1 staining in different images have the same patterns.

Response: Thank you for pointing that out. We apologize for the oversight – we originally wanted to say “*similar* nuclear foci”. We have now included a new panel showing co-staining using HA and DIP1 antibodies in MTD-Gal4>HA-DIP1 ovaries (Supplementary Figure 1q, r). The results show that HA-DIP1 and DIP1 co-localize to the “*same* nuclear foci”.

117 I think it is worthwhile for the authors to explain a little more carefully the disposition of DIP1 staining relative to DAPI and POF to define it as (perhaps) surrounding DNA and chromatin rather than overlapping, or whatever they find to be most accurate for christening this localization pattern.

Response: We have added a sentence as follows:

(page 14, line 4) In general, the staining pattern of DIP1 differs from that of POF. While POF localizes on the chromosomes, DIP1 appears as small foci that surround or decorate the chromosomes.

143 Previously, sisRNAs were carefully described according to numerous criteria. Do the INE-1 sisRNAs meet these criteria or are they just detectable by RT-PCR and in situ? Also, in Fig. 3D “gDNA” is not explained in the legend.

Response: Following the same criteria in our previous study, the INE-1 sisRNAs are defined by their presence in unfertilized eggs by RT-PCR (which contain stable and mature forms of RNAs, without any contamination from pre-mRNAs). In situ hybridization provides evidence that these INE-1 sisRNAs are mostly transcribed from Chromosome 4. We have now amended the legend, “gDNA = genomic DNA”.

250 INE sisRNA “closely associated” with DIP1. Do they co-localize with POF, DAPI to allow a fuller explanation of the relative location of chromatin, RNAs and DIP1 protein?

Response: To describe the localization of INE-1 sisRNAs clearer, we have amended the text as follows:

(page 15, line 17) Fluorescent in situ hybridization showed that INE-1 sisRNAs clustered around and on the fourth chromosomes, although they were also present, to a lesser extent, in other

regions of the nucleus (Fig. 5e). These INE-1 *sisRNAs* are closely associated with the DIP1-positive satellite bodies (some co-localize with DIP1, while other in close proximity to DIP1), suggesting some regulatory relationships (Fig. 5f).

177 I think it is worth explaining prior evidence that RNAi targeting of *sisRNAs*, or this one in particular, is selective, without generally affecting the parent protein-coding transcript (at least in a simple, direct negative way).

Response: We have now mentioned as follows:

(page 6, line 5) We found that manipulating the expression of *sisR-1* using previously reported RNAi lines, which specifically knocked down *sisR-1* but not its parental gene, changed the GSC-niche occupancy.

189 There is some increase in GSC # with the altered *sisR-1* (modify “no increase”)

Response: We amended to “no significant increase” as the increase is not statistically significant.

202 Was it also observed here (as well as in the cited paper) that overexpression of Shg did not increase GSC #? Was the magnitude of increased DE-cadherin caused by Shg and *sisR-1* compared (presumably the former is higher)?

Response: It has been reported previously that overexpression of Shg alone is not sufficient to increase GSC number⁶. Since overexpression of *sisR-1* led to an increase in DE-cadherin (Supplementary Figure 2i, j), and an increase in GSC number (Figure 2h), we hypothesized that *sisR-1* may act through multiple downstream targets other than DE-cadherin alone. This hypothesis led us to test other factors such as Mei-P26. Kindly note that we did not directly compare the levels of DE-cadherin in *sisR-1* overexpression with that of Shg overexpression. Since the Gal4/UAS overexpression system is usually very strong, we assume that the Shg overexpression is presumably higher than that in the *sisR-1* overexpression context, as suggested by the reviewer. We have amended the text to make our point clearer:

(page 7, line 12) However, a previous report showed that overexpression of E-Cadherin alone is not sufficient to increase GSC number, suggesting that *sisR-1* regulates GSCs via cell adhesion and additional mechanisms.

217 Explain prior *ASTR* evidence as an intermediate between *sisR-1* and *rga* better here or earlier.

Response: The new explanation is as follows:

(page 8, line 5) It was shown previously that *sisR-1* represses the abundance of *ASTR*, while *ASTR* functions as a positive regulator of *rga* expression. Therefore, *ASTR* functions as an intermediate between *sisR-1* and *rga* in a negative feedback mechanism.

219 Explain “early & late stage” ovaries of S4C.

Response: Explained in the Figure. Early = before stage 6/7. Late = after stage 6/7.

224 Although *ASTR* RNAi greatly modifies the effect of *sisR-1* RNAi, the converse is also true to some (lesser) degree. This merits a few words about whether this is because null conditions are not achieved or *sisR-1* might have additional targets.

Response: We are not sure if we understood the reviewer's point "the converse is also true" here. If the reviewer is referring to *ASTR* overexpression leading to a subtle decrease in GSC number, it could be true that *sisR-1* may have additional targets, or *ASTR* functions better in cis than in trans. We have added some explanation as follows:

(page 8, line 16) Since the impact of *ASTR* overexpression alone on GSC numbers was milder than that of *sisR-1* RNAi (Fig. 2e and Supplementary Fig. 3e), it suggests that *sisR-1* may have additional targets, or *ASTR* functions better in cis (locally at the *rga* locus) than in trans (when expressed from a transgene).

If we did not get the reviewer's point right, kindly advise again and we will make the appropriate change.

228 I cannot find in ref 36 a description of the *rga/Not2* mutant phenotype. If it is there please cite in a way it might be found but (as will be noted later) the majority of that paper is about other potential partners which have an opposite phenotype. Probably best to omit the reference at this point and discuss the whole thing later.

Response: We agree with the reviewer to discuss it later. We have thus removed the phrase "Consistent with a previous report".

229 It is surprising to me that *Rga* overexpression is less effective than *sisR-1* RNAi at reducing GSC#, raising the question of how much *Rga* protein changes in each case. Really, in situ staining would be needed to answer that question for GSCs but looking at Fig. S4F, G ought to give a clue. However, *Rga* seems reduced by *sisR-1* RNAi! Surely, that is contrary to all other evidence?

Response: There appears to be a misunderstanding that Fig. S4F, G refer to *rga* RNAi, not *sisR-1* RNAi.

267 It is mentioned that *rga* is not on the 4th but in a very subtle way. I think it should be made clearer to readers that the satellite body data are quite distinct from the rest of the functional data. Because of this I find whatever is to be concluded about satellite bodies to be relatively weak. The authors can certainly speculate that there may be similar "bodies" in other locations that are harder to see (and affect *rga*) but it must be clear that it is speculation.

Response: The reviewer share our perspective that *DIP1* regulates *sisR-1* outside the "satellite bodies". To make our story clearer, we have moved Figures 2 and 3 to the back, and discussed our speculation as follows:

(page 18, line 18) We speculate that in the nucleoplasm, DIP1 that does not form observable satellite bodies is sufficient to regulate sisRNAs such as *sisR-1* transcribed from other chromosomes.

I find the most puzzling aspect of the paper to be how results relate to prior studies of CCR4/Not components. I think the sum of past papers could be explained a little more carefully because it is not uniform and contains plenty of information relevant for Shg, Mei-26 mechanisms. Indeed, it does not seem right to just conclude Rga acts independently of other complex members in the light of evidence that those mutations also affect GSCs via DE-cadherin and Mei-26, albeit in an opposite manner.

Response: We appreciate the reviewer's insights. From the paper Yan et al., *Dev Cell* 2014⁷, it appears that knockdown of single components in a complex did not always result in the same GSC phenotypes (for example, the CCR4-NOT complex, splicing complex and ATP synthase). These results suggest a complex crosstalk between individual genes in a complex, possibly with other pathways, so that their phenotypes do not always follow a linear relationship. Knockdown of genes with multiple functions often lead to pleiotropic phenotypes. Moreover, some genes may need to be more carefully fine-tuned as different expression levels may lead to different outcomes. Interestingly, Twin has been proposed to function with distinct partners to mediate different effects on GSC fates⁶. Also a model for protein competition among components of the COP9 complex has been proposed to control GSC differentiation⁸. All these observations point to an idea that the stoichiometry of the components in a complex plays important roles in the cells. Thus, in our context, perhaps the stoichiometry of Rga (NOT2) with the other CCR4-NOT components needs to be fine-tuned to achieve proper GSC number. We have discussed this as follows:

(page 17, line 6) In a large-scale RNAi screen for genes that regulate GSC self-renewal and differentiation, *rga* was identified as a gene required for GSC differentiation. How *rga* regulates GSC self-renewal is currently unknown but our data suggest that GSC-niche adhesion and Mei-P26 are involved. The *rga* gene encodes for the NOT2 protein in the CCR4 deadenylase complex. Surprisingly, studies have shown that other components of the CCR4 complex such as CCR4, Not1 and Not3 function in promoting GSC maintenance. Interestingly, Twin has been proposed to function with distinct partners to mediate different effects on GSC fates⁶. Also, a model for protein competition among components of the COP9 complex has been proposed to control GSC differentiation. All these observations suggest that the stoichiometry of the components in a complex plays important roles in regulating GSC fates. It raises the interesting possibility that the stoichiometry of Rga (NOT2) with the other CCR4-NOT components needs to be carefully fine-tuned to achieve proper GSC number. Studies in yeast had shown that Rga (Not2) exists in functionally independent complexes and mediate other cellular processes such as transcription. Thus, more studies are needed to clarify the molecular mechanism by which Rga regulates GSCs.

Another open question meriting at least speculation is how DIP1 reduces *sisR-1* levels.

Response: This is a very interesting question that we wish to address in the future. We speculate that DIP1 probably binds *sisR-1* and recruits factors that lead to *sisR-1* degradation. Those factors could be RNA degradation machineries such as the exosomes, or RNA modification enzymes that can introduce

modification to sisR-1 to “mark” it for degradation. In future, finding the interacting partners of DIP1 will solve this interesting and important question. We have discussed this as follows:

(page 18, line 19) Since DIP1 is a double-stranded RNA binding protein, it may bind to mature sisRNAs to destabilize them. It may do so by recruiting RNA degradation factors (such as nuclear exosomes) or introducing RNA modification to “mark” sisRNAs for degradation.

The statistics for graphs like Fig. 4Q, 5J... do not seem to be reported appropriately. More words are needed, exact n values should be given and it seems that the asterisks apply always to the number of occurrences of a single number of GSCs, where Fisher’s exact test rather than a t test seems appropriate.

Response: We thank the reviewer for the useful suggestion. We have now added a section on statistical analysis in the methods section, provided exact n in the legends and reanalyzed the data with Fisher’s exact test. The new analyses are consistent with our previous tests.

In summary, I believe that the specific comments above can readily be addressed, as they basically ask for better explanations and a little more extant information. Besides that, I find the evidence supports a story of suitable interest to a wide audience. I will be interested to see if other reviewers have critical comments regarding some of the RNA methodology or CCR4 biology that reflect a deeper expertise.

Response: Once again, we appreciate your interest in our manuscript!

References

- 1 Savva, Y. A. *et al.* RNA editing regulates transposon-mediated heterochromatic gene silencing. *Nature communications* **4**, 2745, doi:10.1038/ncomms3745 (2013).
- 2 Deryusheva, S. & Gall, J. G. Small Cajal body-specific RNAs of *Drosophila* function in the absence of Cajal bodies. *Molecular biology of the cell* **20**, 5250-5259, doi:10.1091/mbc.E09-09-0777 (2009).
- 3 Gardner, E. J., Nizami, Z. F., Talbot, C. C., Jr. & Gall, J. G. Stable intronic sequence RNA (sisRNA), a new class of noncoding RNA from the oocyte nucleus of *Xenopus tropicalis*. *Genes & development* **26**, 2550-2559, doi:10.1101/gad.202184.112 (2012).
- 4 Pek, J. W., Osman, I., Tay, M. L. & Zheng, R. T. Stable intronic sequence RNAs have possible regulatory roles in *Drosophila melanogaster*. *The Journal of cell biology* **211**, 243-251, doi:10.1083/jcb.201507065 (2015).
- 5 Hogan, N. C., Traverse, K. L., Sullivan, D. E. & Pardue, M. L. The nucleus-limited Hsr-omega-n transcript is a polyadenylated RNA with a regulated intranuclear turnover. *The Journal of cell biology* **125**, 21-30 (1994).
- 6 Fu, Z. *et al.* Twin Promotes the Maintenance and Differentiation of Germline Stem Cell Lineage through Modulation of Multiple Pathways. *Cell reports* **13**, 1366-1379, doi:10.1016/j.celrep.2015.10.017 (2015).
- 7 Yan, D. *et al.* A regulatory network of *Drosophila* germline stem cell self-renewal. *Developmental cell* **28**, 459-473, doi:10.1016/j.devcel.2014.01.020 (2014).
- 8 Pan, L. *et al.* Protein competition switches the function of COP9 from self-renewal to differentiation. *Nature* **514**, 233-236, doi:10.1038/nature13562 (2014).

Reviewers' comments:

Reviewer #1 (Remarks to the Author):

I will not describe the main achievements of the paper, since this was done in the review of the original submission. In the previous review, my concerns were mostly structure of the paper rather than technical.

On the whole it has been improved and the authors have addressed most of the reviewers concerns. There are no further technical concerns about the content and presentation of the manuscript in its current form. However, I still think that the story is completed with the first four figures (Figure 1 – 4), and the results shown in the last two figures are another story and not well connected with the first story; first, it seems that satellite bodies have nothing to do with sisR-1, and second, it seems there is no common future in sisRNAs and thus INE-1 sisRNAs could be a totally different class of sisRNAs.

In sum, the major conclusions described in the manuscript represent notable yet incremental advances that will likely be of interest to specialist in the field.

Reviewer #2 (Remarks to the Author):

Regarding responses to my earlier comments (often using the same numbers for line citations as before):

70 ActD effect

The authors' response cites a precedent implying that ActD inhibits turnover of a specific RNA. It seems to me that a hypothesis of this type is necessary here in the text to provide a possible explanation of why sisRNA-1 increases (otherwise careful readers will think the observation and lack of explanation odd).

It is also unfortunate that this observation implies that normal regulation of sisRNA-1 (generation or stability) is disrupted in the presence of actD and hence it is not an ideal test of DIP1 action. More important, (and I think I omitted to raise this previously), it seems there is no basis for concluding that DIP1 acts on sisR-1 stability; it could equally well be acting on sisR-1 generation from rga RNA. I believe that should be explicit, even if the authors express a preference for one hypothesis.

Finally, in a prior publication it was stated that the control actD exp't produces no change in sisR-1 levels (a difference in procedure to be explained?).

Fig.1a,b quantitation . In b is 1.0 in first & third lanes really the same? Does not look possible (all should be on same scale). Numbers not too convincing (small n), especially for (a), where rRNA measure seems hard due to overload.

143 I think it is relevant to point out relative levels of different RNAs. RT-PCR was used for INE-1 sisRNAs because they are much more abundant than parent RNAs but the opposite is true for sisR-1. It would be really useful to have a rough numerical estimate of the relative levels of rga, ASTR and sis-R1 (and INE-1 parent and sisRNA). Previous estimates are that sisR-1 is 1-5% of rga but what about ASTR?

224 My point was that the ASTR RNAi phenotype was significantly altered by sisR-1 RNAi. That implies either that sisR-1 has some significant influence independent of ASTR or that the knockdown of ASTR was partial, so that sisR-1 still significantly influences (overall lower) ASTR levels. Those interpretations should I think be mentioned but the result that the ATR RNAi phenotype IS altered by sisR-1 must be mentioned explicitly (readers will not examine all Figures in detail).

229 Mistaken comment noted, but is increase in Rga protein due to sisR-1 RNAi greater than due to direct RGA over-expression, in keeping with the stronger phenotype of the former?

355-361 (current) It is good that the phenotypes of NOT1 etc. are now mentioned explicitly. However, I find this segment reads like an excuse rather than an attempt to present information and rationalize. I think it is important to say that Twins (say it is CCR4) and other components affect Mei-P26 and Shg in opposite ways to Rga and that these seem to be functional mediators. Those observations narrow the scope for diverse actions and instead focus attention on Rga levels and effects on the NOT/CCR4 complex. I find the comments on Twins additional functions to deflect attention from this central point without offering any specific hypothesis.

Other points:

The revision more clearly separates the rga work from the 4th chromosome satellite bodies. However, both in the Abstract and near the end of the Introduction the transition to DIP1 localization implies that ALL DIP1 acts in these bodies. This mis-impression should be rectified (most simply by explaining that rga is not on the 4th and presumably targeted by DIP1 not in satellite bodies). DIP1 action on sisR-1 outside satellite bodies should also be re-iterated at the end of the Results.

Several grammatical issues (presumably will be dealt with by editors).

Would help to place ASTR RNA on Fig. 1c.

Recommend introducing term "parent gene" early with suitable explanation.

CG8273 described in response as also affecting sisR-1. That should be stated because current text implies that it does not (by saying, in effect, that only DIP1 affected this RNA). It is of interest to know something about specificity- how many sisRNAs affected by DIP1?

From my perspective, the paper as originally presented was appropriate, so the comments above are all intended to help make a final set of changes, rather than as items requiring re-review or precluding publication.

I have also read responses to reviewer 1, who was originally less enthusiastic, and will pass judgment on whether criticisms have been answered satisfactorily.

I note here:

In answer to point 2 it is now said that formation of satellite bodies "may facilitate" RNA decay. It may but there is no evidence here that concentration into a body has a positive or negative effect on function. It is also not clear for INE sisRNA (or sisR-1, as stated earlier) that DIP1 is actually acting on RNA degradation.

In answer to (3) it is useful to have added measurement of some INE "parental genes" but because they are unchanged it seems unreasonable to have text where it is speculated that INE is modulating expression of parent genes. I also find the added speculation about Adar to be unnecessary and confusing.

For (4) I believe both reviewers were concerned how good is the evidence that various RNAi treatments affected one RNA but not another. The author's solution is to quote a past study. That is probably OK but I think one key "finding" is that RNAi directed to intronic sequences does not affect that mRNA. I think that should be repeated here with updated references. Moreover, Fig. 1c or similar should make clear the locations of RNAi constructs used to target rga, ASTR and sisR-1 separately.

The biggest point in the revision that I find problematic is the addition of new data!

All of this new material, concerning sisR-1 action on ASTR reads like a first draft (too many words, awkward language and unbalances the previous emphasis). Moreover, I suspect there are several criticisms to be made (but I am not inclined to make these under the circumstances). My opinion is that these new data should not be in the paper. If reviewer 1 finds them to be sound and instrumental in a positive decision that is fine (but still the authors should revise the text; model 1, then 2, both of which seem odd, then moving to model 3 could be replaced by just giving the results and formulating a preferred model [which I am sure cannot be proved as the only explanation]).

General response: We thank the reviewers again for their time in reviewing our manuscript and the general enthusiasm. Please find our point-by-point response below. Amendments to the manuscript main text are now highlighted in yellow as suggested by the editor.

Reviewer #1 (Remarks to the Author):

I will not describe the main achievements of the paper, since this was done in the review of the original submission. In the previous review, my concerns were mostly structure of the paper rather than technical.

On the whole it has been improved and the authors have addressed most of the reviewers concerns. There are no further technical concerns about the content and presentation of the manuscript in its current form. However, I still think that the story is completed with the first four figures (Figure 1 – 4), and the results shown in the last two figures are another story and not well connected with the first story; first, it seems that satellite bodies have nothing to do with sisR-1, and second, it seems there is no common future in sisRNAs and thus INE-1 sisRNAs could be a totally different class of sisRNAs.

In sum, the major conclusions described in the manuscript represent notable yet incremental advances that will likely be of interest to specialist in the field.

Response: Once again, we appreciate the reviewer's suggestions and enthusiasm in our study. We are very glad that the reviewer is satisfied with our revisions.

Reviewer #2 (Remarks to the Author):

Regarding responses to my earlier comments (often using the same numbers for line citations as before):

70 ActD effect

The authors' response cites a precedent implying that ActD inhibits turnover of a specific RNA. It seems to me that a hypothesis of this type is necessary here in the text to provide a possible explanation of why sisRNA-1 increases (otherwise careful readers will think the observation and lack of explanation odd).

It is also unfortunate that this observation implies that normal regulation of sisRNA-1 (generation or stability) is disrupted in the presence of actD and hence it is not an ideal test of DIP1 action.

More important, (and I think I omitted to raise this previously), it seems there is no basis for concluding that DIP1 acts on sisR-1 stability; it could equally well be acting on sisR-1 generation from rga RNA. I believe that should be explicit, even if the authors express a preference for one hypothesis.

Finally, in a prior publication it was stated that the control actD exp't produces no change in sisR-1 levels (a difference in procedure to be explained?).

Response: We have now added a few sentences describing the possible role of actinomycin D on some nuclear RNA turnover as follows:

(page 5, line 10) It is interesting to note that a previous study had shown that actinomycin D can inhibit the turnover of a nuclear form of *hsr-omega* ncRNA in *Drosophila*²⁸. Together with our observation, it implies that actinomycin D may regulate the decay of specific ncRNAs in the nucleus by a yet unknown mechanism.

The reviewer is correct that DIP1 may regulate stability or generation of *sisR-1*. In the previous version of our manuscript, we have explicitly included this possibility as follows:

(page 5, line 9) This observation suggests that *DIP1* may regulate the stability and/or processing of *sisR-1* in a post-transcriptional manner.

The experiment in the previous publication was performed in a different laboratory using different batches of reagents. The important point of the current actinomycin D experiment is the difference in *sisR-1* stability seen in control versus DIP1 overexpressing ovaries, suggesting a posttranscriptional regulation by DIP1.

Fig.1a,b quantitation . In b is 1.0 in first & third lanes really the same? Does not look possible (all should be on same scale). Numbers not too convincing (small n), especially for (a), where rRNA measure seems hard due to overload.

Response: We have clarified the quantification in the legend as follows:

(page 30, line 8) Numbers below indicate relative band intensities (120 min compared to 60 min) of *sisR-1* normalized to rRNA quantified using ImageJ software from two independent experiments.

All RNAs were quantified using a nanodrop spectrophotometer prior to any experiments to ensure that the loading is equivalent to the best of our ability. We have added a sentence in the Methods section as follows:

(page 22, line 4) RNA was quantified using a Nanodrop spectrophotometer to ensure equivalent loading for subsequent experiments.

143 I think it is relevant to point out relative levels of different RNAs. RT-PCR was used for INE-1 *sisRNAs* because they are much more abundant than parent RNAs but the opposite is true for *sisR-1*. It would be really useful to have a rough numerical estimate of the relative levels of *rga*, *ASTR* and *sisR-1* (and INE-1 parent and *sisRNA*). Previous estimates are that *sisR-1* is 1-5% of *rga* but what about *ASTR*?

Response: The steady-state levels of *sisR-1*, *ASTR* and *rga* are different in during different stages of development and tissues. This can be due to active feedback mechanisms, transcription or other regulatory mechanisms. In ovaries, according to our estimates by northern blotting, *sisR-1*, but not *ASTR*, is readily detectable, suggesting that the relative abundance is *rga*>*sisR-1*>*ASTR*. As for INE-1 *sisRNA*, we added the information in the text as follows:

(page 14, line 20) The *CamKII* *sisRNA* was also previously detected in eggs to be ~10% the level of its parent gene⁵.

224 My point was that the *ASTR* RNAi phenotype was significantly altered by *sisR-1* RNAi. That implies either that *sisR-1* has some significant influence independent of *ASTR* or that the knockdown of *ASTR* was partial, so that *sisR-1* still significantly influences (overall lower) *ASTR* levels. Those interpretations should I think be mentioned but the result that the *ATR* RNAi phenotype IS altered by *sisR-1* must be mentioned explicitly (readers will not examine all Figures in detail).

Response: Thanks for the further explanation. We have discussed as follows:

(page 8, line 20) Since the *ASTR* RNAi phenotype was significantly altered by *sisR-1* RNAi, it also suggests that *sisR-1* may have additional targets or the *ASTR* RNAi was partial.

229 Mistaken comment noted, but is increase in *Rga* protein due to *sisR-1* RNAi greater than due to direct *RGA* over-expression, in keeping with the stronger phenotype of the former?

Response: We believe the *UAS-Gal4* overexpression system is more robust than the *sisR-1* RNAi. In line with the previous response, the stronger phenotype seen in *sisR-1* RNAi could be explained by *sisR-1* having additional target(s).

355-361 (current) It is good that the phenotypes of *NOT1* etc. are now mentioned explicitly. However, I find this segment reads like an excuse rather than an attempt to present information and rationalize. I think it is important to say that *Twins* (say it is *CCR4*) and other components affect *Mei-P26* and *Shg* in opposite ways to *Rga* and that these seem to be functional mediators. Those observations narrow the scope for diverse actions and instead focus attention on *RGA* levels and effects on the *NOT/CCR4* complex. I find the comments on *Twins* additional functions to deflect attention from this central point without offering any specific hypothesis.

Response: We thank the reviewer for the insightful discussion. We have now amended the text as follows to make it more focused:

(page 16, line 18) Interestingly, *Twin* has been proposed to function with distinct partners to mediate different effects on *GSC* fates³⁵. This suggests that other components such as *CCR4* can also have additional functions outside the *CCR4-NOT* deadenylase complex in mediating *GSC* maintenance, thus affecting *GSCs* in opposite ways to *Rga*.

Other points:

The revision more clearly separates the *rga* work from the 4th chromosome satellite bodies. However, both in the Abstract and near the end of the Introduction the transition to *DIP1* localization implies that ALL *DIP1* acts in these bodies. This mis-impression should be rectified (most simply by explaining that *rga* is not on the 4th and presumably targeted by *DIP1* not in satellite bodies). *DIP1* action on *sisR-1* outside satellite bodies should also be re-iterated at the end of the Results.

Response: Following the reviewer's suggestions, we have incorporated the changes as follows:

(page 2, line 10) DIP1 presumably acts outside the satellite bodies to regulate *sisR-1*, which is not on the fourth chromosome.

(page 3, line 17) In this paper, we show that the regulation of a *Drosophila* *sisRNA sisR-1* by DIP1 is important for keeping female germline stem cell homeostasis in place. We also show that DIP1 regulates INE-1 *sisRNAs* and localizes to a previously undescribed nuclear body around the fourth chromosomes, called the satellite body. The regulation of *sisR-1*, which is not on the fourth chromosome, by DIP1 presumably does not occur in the satellite bodies.

(page 15, line 19) Although DIP1 seems to regulate INE-1 *sisRNAs* in the satellite bodies, the regulation of *sisR-1*, which is not on the fourth chromosome, by DIP1 presumably occurs outside the satellite bodies.

Several grammatical issues (presumably will be deal with by editors).

Response: We have proof-read the manuscript once more and corrected the grammatical issues as much as we could.

Would help to place ASTR RNA on Fig. 1c.

Response: Added.

Recommend introducing term "parent gene" early with suitable explanation.

Response: We have now explained further in the introduction as follows:

(page 3, line 5) Studies in *Drosophila* and mammalian cells suggest that *sisRNAs* function in regulating the expression of their parental genes (host genes where they are derived from) via positive or negative feedback loops^{5,10,11}.

CG8273 described in response as also affecting *sisR-1*. That should be stated because current text implies that it does not (by saying, in effect, that only DIP1 affected this RNA). It is of interest to know something about specificity- how many *sisRNAs* affected by DIP1?

Response: We have stated in the text as follows:

(page 4, line 12) The role of CG8273 in *sisR-1* regulation will be reported in a separate study.

As for the number of *sisRNAs* affected by DIP1, it will require deep sequencing of wild type and DIP1 mutants, which is beyond the scope of the current paper.

From my perspective, the paper as originally presented was appropriate, so the comments above are all intended to help make a final set of changes, rather than as items requiring re-review or precluding publication.

Response: We thank the reviewer for the careful reading and constructive feedback throughout the review process. Thanks for your support in publishing this study.

I have also read responses to reviewer 1, who was originally less enthusiastic, and will pass judgment on

whether criticisms have been answered satisfactorily.

I note here:

In answer to point 2 it is now said that formation of satellite bodies “may facilitate” RNA decay. It may be but there is no evidence here that concentration into a body has a positive or negative effect on function. It is also not clear for INE *sisRNA* (or *sisR-1*, as stated earlier) that DIP1 is actually acting on RNA degradation.

Response: We agree. We have indeed discussed in the response previously that the link between satellite body formation and INE-1 regulation needs to be further examined in a separate study.

In answer to (3) it is useful to have added measurement of some INE “parental genes” but because they are unchanged it seems unreasonable to have text where it is speculated that INE is modulating expression of parent genes. I also find the added speculation about Adar to be unnecessary and confusing.

Response: Following the reviewer’s suggestion, we have removed the unnecessary sections in the discussion.

For (4) I believe both reviewers were concerned how good is the evidence that various RNAi treatments affected one RNA but not another. The author’s solution is to quote a past study. That is probably OK but I think one key “finding” is that RNAi directed to intronic sequences does not affect that mRNA. I think that should be repeated here with updated references. Moreover, Fig. 1c or similar should make clear the locations of RNAi constructs used to target *rga*, *ASTR* and *sisR-1* separately.

Response: We have now added the reference here and further clarified in the Methods section as follows:

(page 6, line 10) We found that manipulating the expression of *sisR-1* using previously reported RNAi lines, which specifically knocked down *sisR-1* but not its parental *rga* gene, changed the GSC-niche occupancy⁵ (see Methods).

(page 18, line 15) Note that it has been verified previously that *sisR-1* RNAi knocked down *sisR-1* but not the parental *rga* mRNA⁵.

We have now added the locations of the RNAi constructs in Fig. 1c and explained in the text and legends as follows:

(page 18, line 16) Locations of RNAi constructs were indicated in Fig. 1c.

(page 30, line 13) Locations of RNAi constructs used to target *sisR-1* (red), *ASTR* (green) and *rga* (blue) were indicated.

The biggest point in the revision that I find problematic is the addition of new data!

All of this new material, concerning *sisR-1* action on *ASTR* reads like a first draft (too many words, awkward language and unbalances the previous emphasis). Moreover, I suspect there are several criticisms to be made (but I am not inclined to make these under the circumstances). My opinion is that these new data should not be in the paper. If reviewer 1 finds them to be sound and instrumental in a positive decision that is fine (but still the authors should revise the text; model 1, then 2, both of which

seem odd, then moving to model 3 could be replaced by just giving the results and formulating a preferred model [which I am sure cannot be proved as the only explanation]).

Response: Since reviewer 1 is positive with the newly added data, we will keep it in the manuscript. We have now revised the text to make it more streamlined and focused in accordance to the reviewer's recommendation. Please see highlighted text for details (pages 10 and 11).